# LGR5 controls extracellular matrix production by stem cells in the developing intestine

Valeria Fernandez Vallone[†], Morgane Leprovots, Didac Ribatallada-Soriano, Romain Gerbier, Anne Lefort, Frédérick Libert, Gilbert Vassart & Marie-Isabelle Garcia[*] (iD)

## Abstract

The Lgr5 receptor is a marker of intestinal stem cells (ISCs) that regulates Wnt/b-catenin signaling. In this study, phenotype analysis of knockin/knockout Lgr5-eGFP-IRES-Cre and Lgr5-DTReGFP embryos reveals that Lgr5 deficiency during Wnt-mediated cytodifferentiation results in amplification of ISCs and early differentiation into Paneth cells, which can be counteracted by *in utero* treatment with the Wnt inhibitor LGK974. Conditional ablation of Lgr5 postnatally, but not in adults, alters stem cell fate toward the Paneth lineage. Together, these *in vivo* studies suggest that Lgr5 is part of a feedback loop to adjust the Wnt tone in ISCs. Moreover, transcriptome analyses reveal that Lgr5 controls fetal ISC maturation associated with acquisition of a definitive stable epithelial phenotype, as well as the capacity of ISCs to generate their own extracellular matrix. Finally, using the *ex vivo* culture system, evidences are provided that Lgr5 antagonizes the Rspondin 2-Wnt-mediated response in ISCs in organoids, revealing a sophisticated regulatory process for Wnt signaling in ISCs.

**Keywords** development; Lgr5; organoids; Rspondins; stem cells
**Subject Categories** Development & Differentiation; Signal Transduction; Stem Cells & Regenerative Medicine

## Introduction

The adult intestinal epithelium is a specialized tissue involved in nutrient absorption and protection against pathogens and environmental toxic agents. Under homeostatic conditions, within few days, this epithelium undergoes rapid and constant renewal supported by a pool of intestinal stem cells (ISCs), also called crypt base columnar cells, identified by the expression of the Lgr5 receptor [1]. Restricted to the bottom of the crypts of Lieberkühn, ISCs have the capacity to both self-renew and give rise to transit-amplifying cells, which differentiate along the villus architecture into all the cell lineages of the epithelium, i.e., absorptive enterocytes, mucus-producing goblet cells, hormone-secreting enteroendocrine cells, Paneth cells generating antimicrobial products, and the type 2 immune response-inducer tuft cells [2]. Other populations of slowly cycling or label-retaining reserve stem cells have been identified to efficiently regenerate the intestinal epithelium upon loss of Lgr5-expressing stem cells; additional evidences have been provided for coexistence and possible mutual interconversion between these two stem cell populations [3–5]. However, the molecular mechanisms associated with emergence and establishment of adult stem cells during development still remain incompletely understood. By using transgenic mouse lines, evidences have been provided that Cdx2 is a master transcription factor required for intestinal specification before the embryonic stage E14 [6,7]. Thereafter, the intestinal epithelium undergoes a profound remodeling, in part instructed by the underlying mesenchyme, leading to appearance of separate domains constituted by villus and intermingled intervillus regions [8–10]. Coherent with a proximal-to-distal wave of cytodifferentiation along the intestine mediated by the Wnt/b-catenin pathway around E14.5, the Wnt/β-catenin target gene Lgr5 becomes upregulated and identifies cells (ISC precursors) restricted to the intervillus regions that grow as adult-type organoids in the *ex vivo* culture system [11–14]. After birth, concomitant with Paneth cell lineage differentiation, intestinal crypts will be formed by invagination of the intervillus regions into the surrounding mesenchyme, bearing in their bottom the Lgr5-expressing adult ISCs? [15].

Despite general consensus on the function of the Lgr5 receptor as a Wnt/β-catenin signaling modulator in stem cells, how it does so remains still controversial. First of all, *in vitro*, binding of the natural ligands Rspondins to the receptor Lgr5 has been demonstrated to either enhance or inhibit the Wnt pathway depending on the cell type analyzed [16–20]. Secondly, *in vivo*, homozygous *Lgr5-LacZNeo* knockin/knockout embryos deficient for Lgr5 exhibited an overactivated Wnt/b-catenin signaling pathway at birth associated with precocious Paneth cell differentiation, this suggesting a negative regulatory function of Lgr5 on this cascade [21]. However, conditional ablation of the Lgr5 function in adults did not result in significant alteration in Paneth cell differentiation [17]. Moreover, the molecular mechanisms associated with Lgr5 function in ISCs are still debated, does this G-protein-coupled receptor simply control Wnt signaling at the extracellular level by trapping the E3 ubiquitin

Faculty of Medicine, Institut de Recherche Interdisciplinaire en Biologie Humaine et Moléculaire (IRIBHM), Université Libre de Bruxelles ULB, Brussels, Belgium
*Corresponding author. Tel: +32 2 555 4195; Fax: +32 2 555 4655; E-mail: mgarcia@ulb.ac.be
†Present address: 1 Charité – Universitätsmedizin Berlin, Berlin Institute of Health (BIH), Berlin, Germany

ligase Znrf3/Rnf43 at the cell membrane, or does Lgr5 signal via its transmembrane domains and intracellular tail [17,22,23].

In the present report, we further investigated the role of the Lgr5 receptor during intestinal development by analyzing the transcriptome of Lgr5-expressing or Lgr5-deficient ISCs just after the onset of the Wnt-mediated cytodifferentiation (E16) and in adult homeostatic tissues. We provided evidences that Lgr5 controls ISC maturation associated with acquisition of a definitive stable epithelial phenotype, as well as the capacity of ISCs to generate their own extracellular matrix. In addition, using the *ex vivo* culture system, we demonstrate that the Lgr5 receptor/Rspondin 2 ligand interaction negatively regulates the pool of ISCs in organoids, in a process associated with modulation of epithelial extracellular matrix production.

# Results

### *In utero* inhibition of Wnt activity counteracts premature Paneth cell differentiation induced by Lgr5 deficiency in the intestine

To clarify the molecular function of the Lgr5 ISC marker in the embryonic intestine, we investigated the potential phenotype of knockin/knockout (KO) homozygous Lgr5 embryos from the Lgr5-GFP-Cre$^{ERT2}$ and Lgr5-DTReGFP mouse strains [1,24]. Since Lgr5 KOs generated from both transgenic lines show neonatal lethality associated with ankyloglossia, histological analyses were performed at E18.5 (Fig EV1A). Despite no evidence of gross architectural epithelial alterations, Lgr5 KOs exhibited early differentiation toward the Paneth lineage as revealed by Lendrum's staining (that evidences Paneth cell granules) as well as qRT–PCR analysis of E18.5 tissues (Figs 1A and B, and EV1B, Table EV1). In addition, Lgr5 KOs showed fourfold increased expression of Wnt/β-catenin target genes (*Ascl2, Axin2*), histologically detected in the intervillus (IV) regions as compared to wild types (WTs) (Fig 1C and D). This was associated with an early expansion of the eGFP-positive (+ve) stem cell pool both at E16.5 and E18.5 in Lgr5 KO embryos as compared to heterozygous (HEs) (Fig 1E). Of relevance, upregulation of the truncated *Lgr5* transcript itself was even higher [10-fold versus (vs) WTs], suggesting a negative control of the Lgr5 receptor on its own expression (Fig 1D). Altogether, these data confirm previous studies on other Lgr5-deficient mouse strains [21,25] and suggest that Lgr5 deficiency generates overactivation of the Wnt/β-catenin pathway in the prenatal small intestine inducing an expansion of ISC precursors and leading to premature Paneth cell differentiation around birth. ISCs co-express the two paralogue receptors Lgr4 and Lgr5 [17,26]. Since deficiency for the Lgr4 receptor leads to ISC loss due to insufficient Wnt signaling in cultured crypts, we assessed the long-term growth properties of Lgr5-deficient ISCs in the *ex vivo* culture system [26]. Irrespective of the mouse strain of origin, upon initial seeding, Lgr5 KO E18.5 small intestines generated a threefold to fourfold increase in the absolute number of growing organoids, which exhibited higher complexity as compared to WTs and HEs (Figs 1F and EV1C). As reported earlier, such higher organoid complexity could be explained by the presence of Paneth cells in Lgr5 KO versus control samples at the time of seeding [14]. The stemness status of Lgr5 KO ISCs was studied by replating Lgr5-DTReGFP samples for more than 20 passages (Fig EV1D and E).

Organoid growth and Wnt target gene expression were maintained over passages in Lgr5 KOs demonstrating that long-term replating of Lgr5 KO organoids is preserved *ex vivo*. However, the Wnt tone in Lgr5 KO organoids was not sufficient to confer reduced growth requirements as compared to WTs since KO organoids remained dependent on Rspo1 to grow *ex vivo* (Fig 1G).

In attempts to reduce *in vivo* the excessive Wnt signaling tone observed in Lgr5 KO embryos, we treated pregnant females (Lgr5-DTReGFP and Lgr5-GFP-Cre$^{ERT2}$ strains) with the orally administrable Wnt inhibitor LGK974. This inhibitor of the acyl transferase Porcupine (which alters Wnt ligand secretion) restores normal Wnt levels in tumor-bearing mice without affecting highly proliferative tissues such as the intestine [27]. Pilot experiments demonstrated that the compound efficiently crosses the placenta but that its administration before embryonic stage E11.5 can affect normal embryonic development (Fig EV2A). Then, we tested two different administration windows for daily oral gavage (dose of 3 mg/kg/day), i.e., starting before (E13-E15) or during (E15-E17) the onset of Wnt-mediated cytodifferentiation (Fig 2A). Administration of LGK974 between E15-E17, but not earlier, reduced Paneth cell differentiation in Lgr5 KOs as compared to control levels (Figs 2B and EV2B). The treatment reduced expression of Wnt target genes (*Axin2, Ascl2, Lgr5 ex1*) and the ISC pool (as quantified with the Olfm4 marker) within the IV region without significantly altering expression of other reported stem cell markers (*Hopx, Tert*) (Figs 2C–E and EV2C). LGK974 administration also induced down-regulation of the Paneth cell marker *Defa5* but not that of other cell lineages markers (Figs 2E and EV2C and D). In a converse series of experiments, attempts to upregulate Wnt/β-catenin signaling in *Lgr5*-expressing stem cells between E15-E17 by a genetic approach via deletion of the β-catenin exon 3 (encoding the sequences targeting the protein for proteasome degradation [28]) exacerbated the Paneth cell differentiation phenotype in Lgr5 KO embryos as compared to Lgr5 HEs (2.29 ± 0.33 versus 1.15 ± 0.22 Paneth cells/10 intervilli, respectively; *P* = 0.0317) (Fig EV2E). Together, these rescue experiments further strengthened the notion that the Lgr5 receptor is involved in negative regulation of the Wnt/β-catenin activity at the onset of cytodifferentiation in the embryonic intestine.

### Postnatal Lgr5 ablation in ISCs alters stem cell fate towards the Paneth cell lineage

To determine whether the phenotype observed in Lgr5-deficient embryos could be reproduced postnatally (PN) when the Paneth cells normally emerge in control tissues, we generated conditional deficient-Lgr5 mice (cKO). These mice are double heterozygous Lgr5$^{GFP-CreERT2/flox}$ in which cre-mediated deletion of Lgr5 exon 16 causes a frameshift and a null phenotype [1,17]. Following 3 consecutive tamoxifen injections to lactating females (PN days 6–8), we compared the fate of cKO Tom-recombined cells to that of control heterozygous (HE) Lgr5$^{GFP-CreERT2/+}$/Rosa26R-Tom littermates after 10 days of chase. No significant differences in terms of clone number were observed in Lgr5-ablated Tom$^{+ve}$ tissues as compared to controls, suggesting that stemness was preserved during this period of chase (Fig 3A and B). However, cKO PN18 Lgr5-ablated Tom$^{+ve}$ tissues exhibited a clear bias toward Paneth cell differentiation (Fig 3C). Such phenotype was observed in proximal and distal small intestines

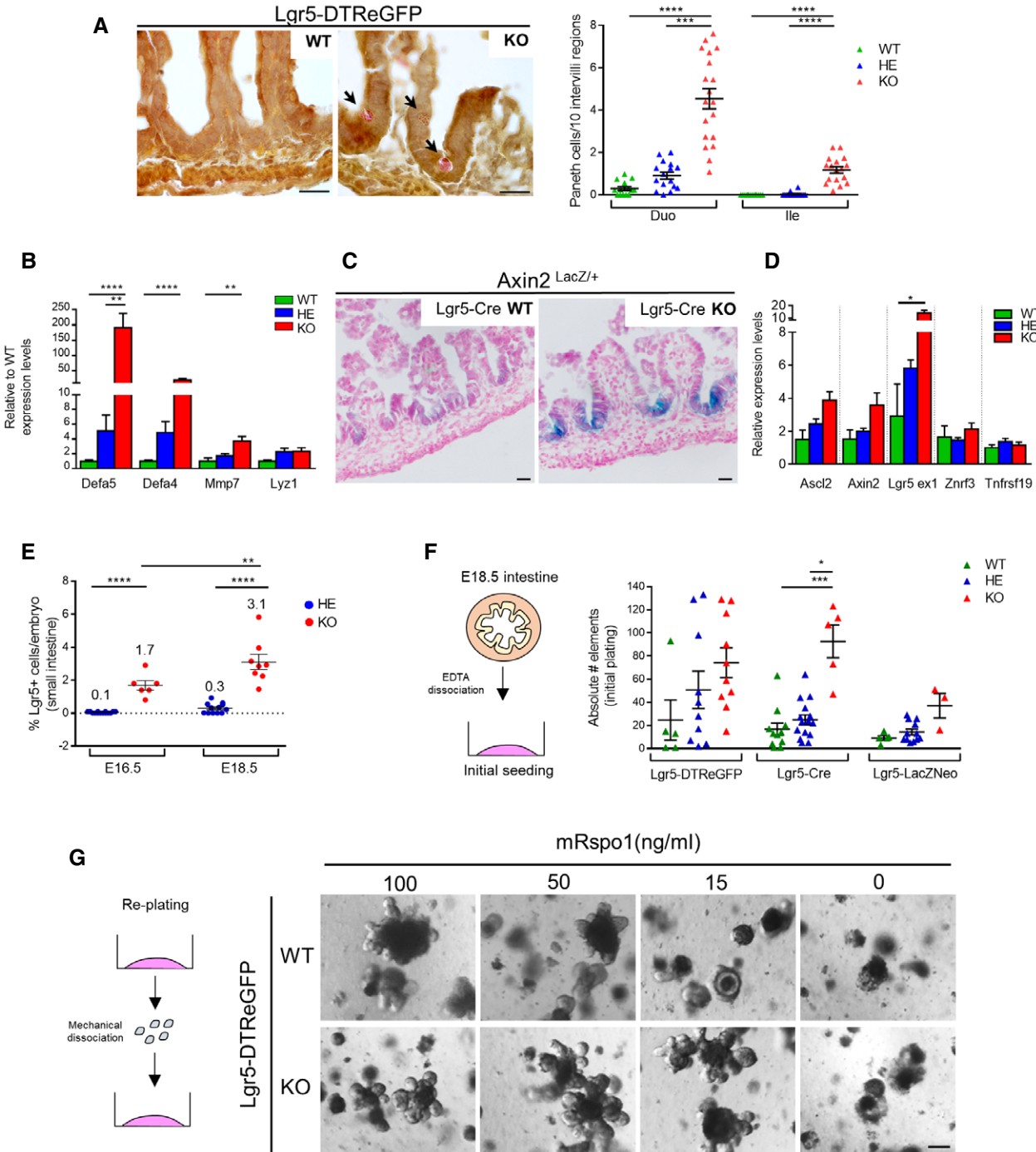

**Figure 1. Lgr5 deficiency induces early Paneth cell differentiation and stem cell expansion in the small intestine at E18.5.**

A   Paneth cell quantification on Lgr5-DTReGFP duodenum (Duo) and ileum (Ile). Left panel: Representative images of Lendrum's staining that evidence Paneth cell granules. Arrows show differentiated Paneth cells. Right panel: Quantification of the Paneth cells per 10 intervillus regions. Each symbol indicates the value for a given embryo.

B   Expression analysis by qRT–PCR of the indicated Paneth cell markers in Lgr5-DTReGFP ileums (*n* = 8 WT, 12 HE, 10 KO).

C   X-gal staining in Axin2^Lac/+^-Lgr5-GFP-CreERT2 WT or KO ileums.

D   Gene expression analysis by qRT–PCR of stem cell markers in Lgr5-DTReGFP ileums (*n* = 3 WT, 6 HE, 6 KO).

E   FACS quantification of the number of eGFP$^{+ve}$ (ISC) cells per small intestine at the indicated developmental stages in Lgr5 HEs and Lgr5 KOs. Each symbol indicates the value for a given embryo. Mean value for each group is depicted on the graph.

F   Plating efficiency of ex vivo cultured E18.5 small intestines quantified 6 days after initial seeding. Each symbol indicates the value for a given embryo.

G   Influence of mouse Rspondin 1 concentration on growth of replated Lgr5-DTReGFP WT and Lgr5 KO organoids after 5 days of culture.

Data information: Scale bars, 20 μm (A, C) and 50 μm (G). Data are represented as means ± SEM. *$P$ < 0.05; **$P$ < 0.01, ***$P$ < 0.001; ****$P$ < 0.0001 by Kruskal–Wallis test followed by Dunn's multiple comparison test (A, B, D, F) and two-way ANOVA followed by Tukey's multiple comparisons test (E).

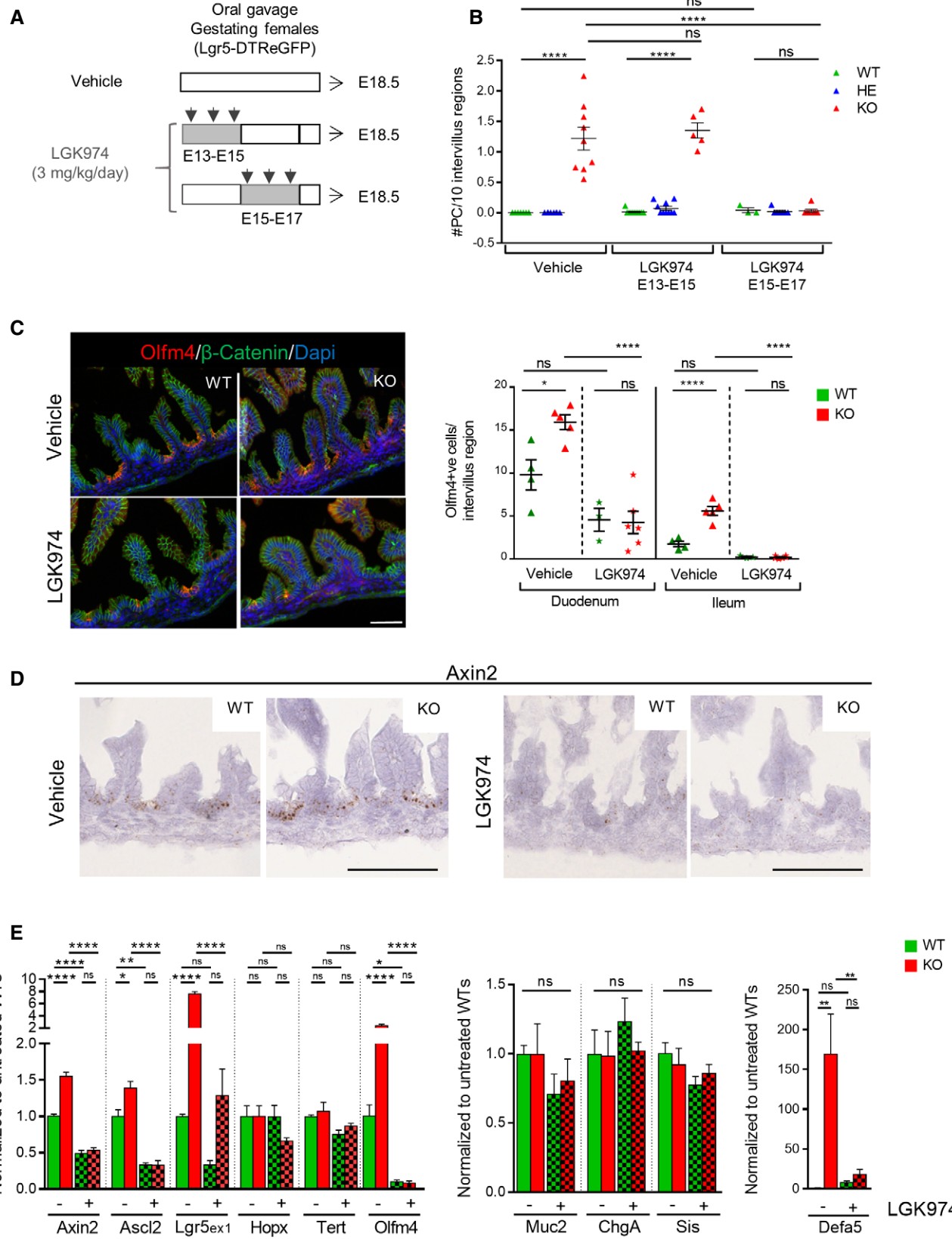

**Figure 2.**

Figure 2. **In utero inhibition of Wnt activity counteracts early Paneth cell differentiation induced by Lgr5 deficiency.**

A  Design of the experiment. Gestating Lgr5-DTReGFP females were vehicle- or LGK974-treated by oral gavage between E13-E15 or E15-E17. Small intestines of treated embryos were analyzed at E18.5.
B  Paneth cell quantification on Lgr5-DTReGFP ileums. Each symbol indicates the value for a given embryo.
C  Immunofluorescence showing Olfm4$^{+ve}$ cells in Lgr5-DTReGFP duodenums. Cell membranes are shown with b-catenin, and nuclei were counterstained with DAPI. Quantification of Olfm4$^{+ve}$ cells per intervillus regions in duodenum and ileum. Each symbol indicates the value for a given embryo.
D  Axin2 expression levels detected by RNAscope on Lgr5-DTReGFP proximal intestine.
E  Gene expression analysis by qRT–PCR of stem cell and differentiation markers in Lgr5-DTReGFP ileums (vehicle-treated: 7 WT and 7 KO; LGK974-treated: 3 WT and 7 KO).

Data information: scale bars, 50 μm (C) and 100 μm (D). Data are represented as means ± SEM. ns not significant, *$P < 0.05$; **$P < 0.01$; ****$P < 0.0001$ by two-way ANOVA followed by Tukey's multiple comparisons test (B, C, E).

(Fig EV3A and B). When Lgr5 ablation was induced in adults already bearing a definitive number of Paneth cells in crypts, such phenotype was not observed, and no significant differences were detected in the proportion of traced clones (Fig 3C). Together, these data are consistent with the fact that loss of Lgr5 function in homeostatic adult tissues is not associated with an overt phenotype [17] whereas the absence of this receptor during prenatal and early postnatal stages impacts on stem cell fate.

### Lgr5 controls extracellular matrix autocrine production in stem cells

To investigate the molecular pathways altered by Lgr5 deficiency, we analyzed the transcriptomic profile of fluorescence-activated cell sorting (FACS)-isolated GFP$^{+ve}$ ISC precursors from Lgr5-DTReGFP HEs and KOs at E16.5 [2 independent pools of Lgr5 HEs ($n = 7$ and 8) and Lgr5 KOs ($n = 2$ and 6)] (Dataset EV1). At this developmental stage, potential impact of Paneth cells was excluded based on Lendrum's stainings. Moreover, the gating strategy based on doublets exclusion further limited potential contamination of sorted ISCs with any other cell type (Fig EV4A). RNAseq on these samples identified 487 differentially expressed genes (96 upregulated/391 downregulated) in Lgr5 KOs versus HEs (false discovery rate FDR 0.1 and fold change FC above 1.5) (Fig 4A and B). Lgr5 deficiency was strikingly associated with de-enrichment in the Epithelial–Mesenchymal Transition Gene dataset (47 genes modulated out of 200 genes in the EMT dataset, $P$ value $4.e^{-60}$) reminiscent of profound reorganization of the matrisome with significant reduction in extracellular matrix (ECM) structural constituents, including collagen fibrils, in Lgr5-deficient ISC precursors (Fig 4B and C). In accordance with ECM playing a role in development [29], these downregulated genes were also associated with tissue development, morphogenesis, and regulation of cell migration (Fig 4C). Coherent with evolution of precursor ISCs toward a more mature stage over time, epithelial expression of ECM components was detected in fetal but not in adult tissues (Fig 4C and D, Table EV1). Note that decreased expression of ECM markers in Lgr5 E16.5 KOs as compared to WTs could not be visualized by RNAscope, likely due to low expression levels in a limited pool of ISCs. To further test this hypothesis, we compared the transcriptomes of E18.5 embryos and adult Lgr5 HE ISCs (Dataset EV2). Stem cell maturation was associated with downregulation of EMT processes with decreased expression of ECM-associated genes (Figs 4B–D and EV4B). Moreover, in agreement with the study of Navis et al [30] investigating transcriptomic changes during suckling-to-weaning transition, maturation also involved metabolic changes in ISCs (Fig EV4B).

Transition from a mesenchymal to an epithelial phenotype is controlled by various signaling pathways [31]. Consistent with an involvement of the Wnt/β-catenin pathway, Lgr5 E16.5 KO GFP$^{+ve}$ ISCs exhibited upregulation of the *Axin2*, *Bax*, and *Edn1* target genes (Fig 4A). While relative levels of the most expressed Wnt Frizzled (Fzd5 and Fzd7) and Lrp co-receptors (Lrp6) were not significantly changed by the developmental stage of ISCs or the Lgr5 expression level (HE or KO), expression of Wnt ligands varied over time (Fig EV4C and D). Indeed, expression of the non-canonical Wnt5a ligand, predominant in HE ISCs at E16.5 (in agreement with data from [13]), progressively dropped at a later prenatal stage (E18.5) and adulthood whereas expression of the canonical Wnt3 ligand increased in ISCs during the same period (Fig EV4C–E). In Lgr5-deficient ISCs, the Wnt5a-to-Wnt3 shift expression was detected at an earlier stage (E16.5) as compared to Lgr5 HEs pairs (Fig EV4D). In addition, the TGFβ pathway involved in EMT was found deregulated in the absence of the Lgr5 receptor with downregulation of *Bmp3* and *TGFβi* in E16.5 KOs as compared to HEs (Fig 4A). Moreover, whereas the TNFα/NFkB and IFNα/γ pathways involved in the inflammatory response were found upregulated during normal ISC maturation, these pathways, well known to crosstalk with the Wnt cascade in a complex way, appeared downregulated in E16.5 Lgr5-null ISCs as compared to control E16.5 ISC HEs [32,33] (Fig 4A and B). Surprisingly, when Lgr5 function was specifically disrupted in adults using Lgr5$^{Cre/flox}$ mice, most of these pathways were instead upregulated (Fig 4B, Dataset EV3). As an exception, the IFN α/γ pathway appeared similarly downregulated in Lgr5-deficient embryo or adult stem cells (Fig 4B). Altogether, these data indicated that the transcriptome of ISC significantly changes from prenatal to adult stage, acquiring a definitive epithelial phenotype associated with a net decrease in the capacity to generate its own ECM niche and an increased ability to respond to inflammatory signals. Moreover, the absence of Lgr5 expression in the early ISC precursors (i.e., just after their emergence in intervillus regions) leads to accelerated conversion of ISCs toward their epithelial mature phenotype whereas acute ablation of Lgr5 in adults rather shifts epithelial stem cells toward a mesenchymal-like phenotype. Additional studies will be needed to determine whether ISC maturation in Lgr5-null embryos is not only accelerated but also improperly executed.

### Rspondin 2 ligand/ Lgr5 receptor interaction regulates stem cell fate in organoids *ex vivo*

Rspondins (Rspo1/Rspo2/Rspo3/Rspo4) have been reported to behave as redundant ligands for Lgr receptors, leading to enhanced

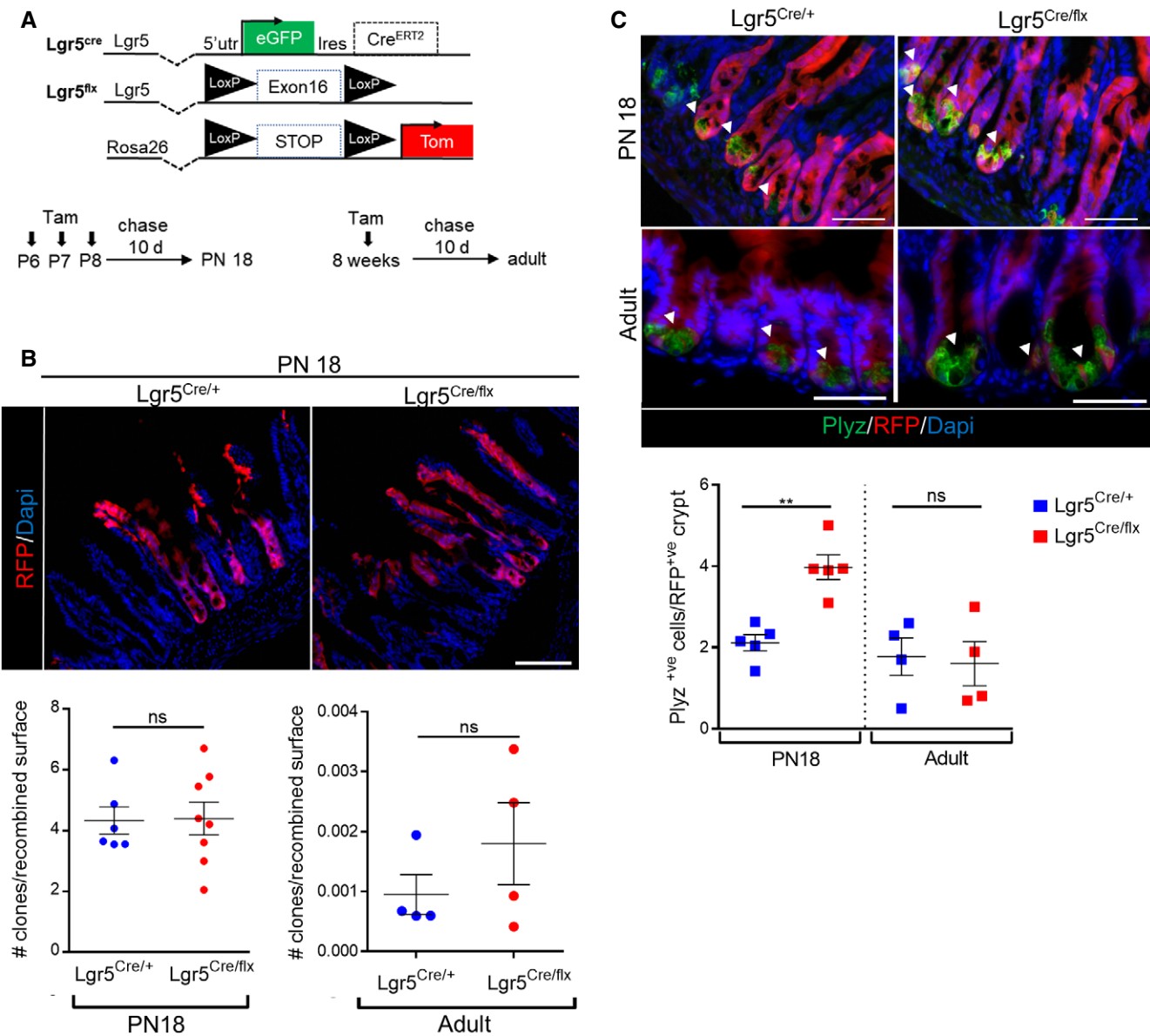

**Figure 3. Postnatal Lgr5 ablation in ISCs alters stem cell fate toward the Paneth cell lineage.**

A  Schematic representation of the genetic elements for lineage tracing and Lgr5 ablation during postnatal development and adult homeostasis and experimental design. Lactating females were tamoxifen (Tam)-treated between PN 6 and PN 8. Small intestines of treated pups were analyzed 10 days after the last injection (PN 18); adult mice received one tamoxifen intra-peritoneal injection, and intestines were analyzed 10 days later.

B  Representative pictures of lineage tracing in Lgr5-expressing or Lgr5 cKO intestine after 10 days of chase and quantification of the number of traced clones per recombined surface at PN18 (ileum). Each symbol indicates the value for a given mouse.

C  Representative immunofluorescence pictures showing Paneth cells (Plyz+ve) in traced clones (RFP+ve) from control (Lgr5Cre/+) and cKO (Lgr5Cre/flox) PN18 and adult ileums. Arrowheads point to Plyz+ve cells in traced clones. Quantification of the number of Paneth cell number per recombined RFP+ve crypt on control and cKO ileums in PN18. Each symbol indicates the value for a given mouse.

Data information: Scale bars: 100 μm (B) and 50 μm (C). Data are represented as means ± SEM. ns, not significant, **$P < 0.01$ versus controls by unpaired *t*-test (B left −PN18) Mann–Whitney test (B right—adult, C).

Wnt/β-catenin activity [16–20]. We investigated their gene expression profile during intestinal maturation by combined qRT–PCR and *in situ* hybridization experiments. Overall, all Rspondins were expressed diffusely at very low levels in the epithelium (Figs 5 and EV5A, Table EV1). However, Rspo2 and Rspo3 appeared as the most expressed members of the family, being especially

detected in stromal cells located in proximity to the Lgr5+ve ISCs in fetal and adult tissues (Fig 5). Expression of the Lgr5 receptor progressively increased during development, and its expression became restricted to the IV region in late fetal stages and in adult crypts afterward (Fig 5). Thus, we studied the impact of each Rspondin member on Lgr5-DTReGFP WTs or KOs organoid growth

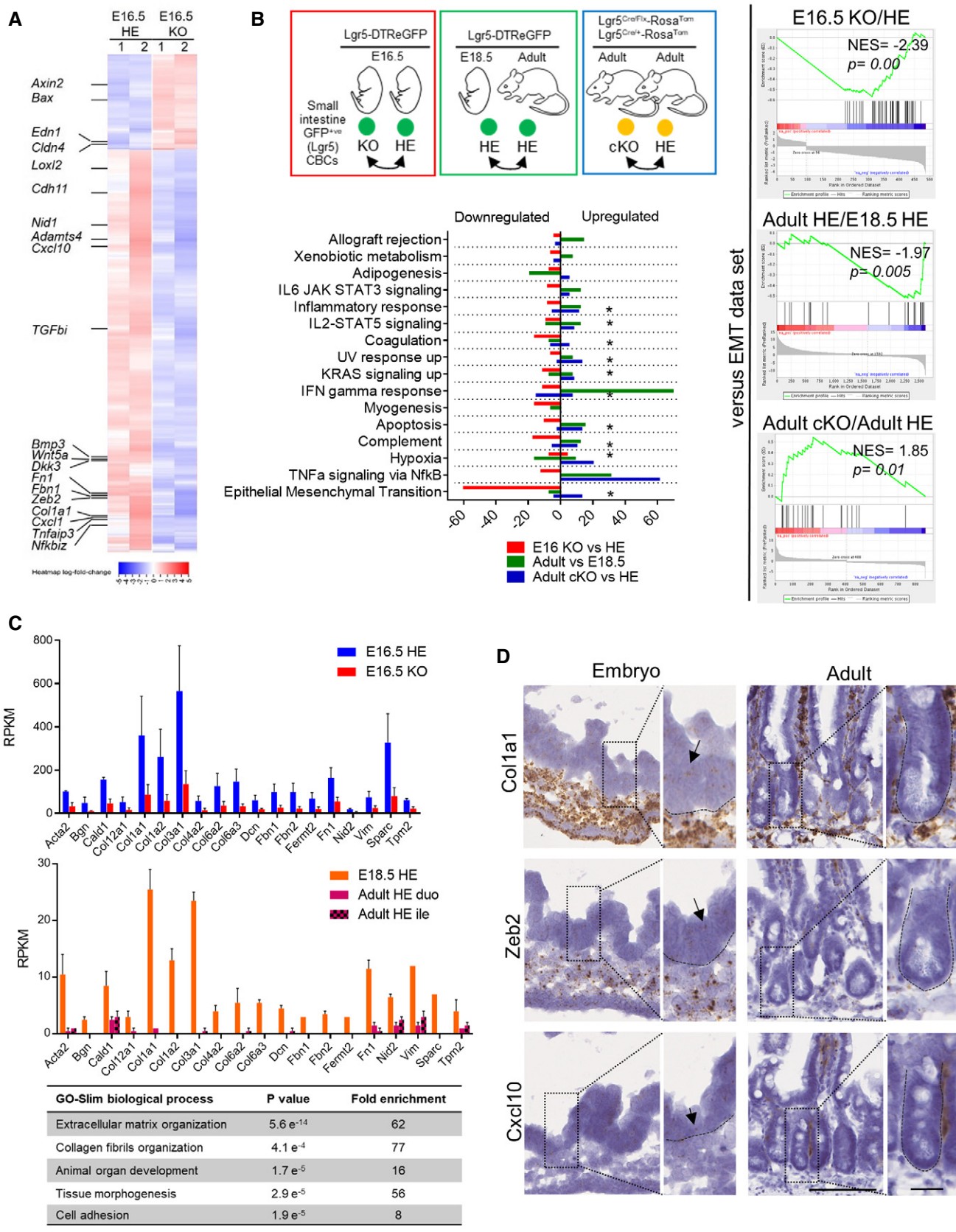

**Figure 4.**

**Figure 4. Transcriptome analysis of Lgr5-deficient ISC precursors.**

A   ISC (eGFP$^{+ve}$) cells were FACS-sorted from Lgr5-DTReGFP embryos at E16.5 and subjected to RNAseq analysis. Heatmap of differentially regulated genes in HE and KO ISCs at E16.5.

B   Compared Hallmarks for upregulated and downregulated genes in the transcriptome of E16.5 Lgr5 KOs versus E16.5 HEs, adult HEs versus E18.5 HE embryos, and adult Lgr5-Cre cKO versus HEs. Asterisks refer to given pathways for which differentially modulated genes were found upregulated and downregulated. Right panels: GSEA showing enrichment of the epithelial to mesenchymal transition (EMT) dataset in E16.5 Lgr5 KOs versus E16.5 HEs, adult HEs versus E18.5 HE embryos, and adult Lgr5 cKO versus adult HE modulated genes. NES: Normalized Enrichment Score.

C   Expression levels of genes commonly downregulated in ISCs of Lgr5 KOs versus Lgr5 HEs at E16.5 and Lgr5 adult versus E18.5 HEs. RPKM: reads per kilobase of transcript per million mapped reads. Below: Gene Ontology analysis of downregulated genes identified in the Epithelial–Mesenchymal Transition Gene Set (Hallmark MSigDB collection).

D   Expression profile of Col1a1, Zeb2, and Cxcl10 genes detected by RNAscope in embryonic and adult intestines. Insets with higher magnification are depicted on the right of each corresponding picture, boundaries between epithelium and mesenchyme are shown with dotted lines, and arrows point to epithelial expression.

Data information: Scale bars, 100 and 20 μm for the insets. Biological replicates for RNAseq experiments on ISC (A, B, C): Lgr5-DTReGFP E16.5 independent embryonic pools, each obtained from independent litters (*n* = 2 HE, 2 KO); Lgr5-DTReGFP E18.5 individual embryos (*n* = 2 HE) and Lgr5-DTReGFP adult individual animals (*n* = 2 HE); and Lgr5-Cre/flox adult individual animals (*n* = 2 HE, 3 cKO). Data are represented as means ± SEM (C).

*ex vivo* after replating (Fig 6A). Organoid growth was substantially affected by Rspo concentration and type; Rspo4 was associated with low organoid survival at the highest concentration (Fig 6A and B). Then, organoid complexity was analyzed at the minimum Rspondin concentration tested allowing 100% survival, this corresponding to 100 ng/ml for Rspo1 and Rspo3 and 5 ng/ml for Rspo2 (Fig 6B). Similar branching coefficients were detected between WT and KO replated organoids in Rspo1 or Rspo3 conditions, a difference as compared to experiments performed upon initial seeding on WT and KO samples with Rspo1 (those in Figs 1F and EV1C). This could be at least explained by the fact that WT samples directly isolated from tissues or after subculture do not bear the same Paneth cell content. In contrast, subculture of replated organoids in the presence of Rspo2, though associated with similar survival rates, showed significant reduction in organoid branching in Lgr5 KOs as compared to WTs in the presence of Rspo2 (Fig 6B). This suggested that Rspo2 positively regulates crypt morphogenesis or branching in a Lgr5-dependent way. To further investigate this phenotype, RNAseq followed by Degust (FDR 0.001, FC above 16) and MSigDB analyses were performed on organoid samples (Dataset EV4). These *in silico* analyses revealed significant enrichment in the C2 Curated gene set "Sansom APC targets" (*P* value 1.5.e$^{-24}$) (Fig 6C). In Lgr5 WTs, all ligands (including Rspo2) potentiated Wnt activity in a dose-dependent manner, i.e., higher concentration of ligand correlated with increased Wnt tone, defined by expression of Wnt target genes, expression of ISCs markers, and differentiation toward the Paneth cell lineage (Fig 6C–E). Similar Rspo1 and Rspo3 dose-dependent effects on Wnt activity were detected in Lgr5 KO organoids (Fig 6C and D). In contrast, in Rspo2 cultures, expression of Wnt target genes, and Paneth cell markers remained at higher levels in Lgr5 KOs as compared to Lgr5-expressing organoids, irrespective of the Rspo2 concentration (Figs 6C and D, and EV5B and 5C and D). In line with these transcriptomic data, the number of Olfm4$^{+ve}$ ISCs appeared significantly increased in Lgr5 KO organoid protrusions as compared to WTs in Rspo2-containing medium (Fig 6E). In addition, Paneth cell differentiation, known to be induced by high Wnt tone, appeared to be negatively regulated by Rspo2/Lgr5 interaction (Figs 6D and EV5D). Together, these data suggested that the Lgr5 receptor acts as an inhibitor of the Rspo2 response in ISCs. This is compatible with the observed reduced branching and high Wnt activity detected in Rspo2-replated Lgr5-null organoids. Indeed, a high Wnt tone induced by Apc loss or

Wnt3a ligand supplementation promotes cyst morphology associated with reduced organoid branching [17]. Though, it is not excluded that crypt morphogenesis regulated by the Lgr5/Rspo2 interaction might also involve Wnt-independent pathways. Moreover, RNAseq analysis also revealed that Rspondins regulate matrisome component expression in organoid cultures (canonical pathways significantly enriched in the "Naba Matrisome" gene dataset, *P* value 6.e$^{-24}$) (Fig EV5E). In Lgr5 KO organoids, matrisome expression appeared dependent on Rspo1 and Rspo3 concentrations but remained instead insensitive to Rspo2 concentrations (as observed for genes like *Lama5*, *Lamb2*, *Col4a1*, *Col4a2*, *Loxl4*). Taken together, these observations suggest a specific role of the Rspo2/Lgr5 axis to regulate ISC behavior that would involve, in part, modulation of ECM production by these cells.

## Discussion

Since the identification of the atypical GPCR receptor Lgr5 as a bone-fide stem cell marker in the normal intestinal epithelium as well as in cancer cells, the potential biological function of this receptor has been questioned but still, it remains elusive and controverted [34]. The biological function of Lgr5 was initially addressed in Lgr5-null embryos using the Lgr5-LacZNeo mouse line, in which exon 18 coding for the transmembrane and intracellular tail is replaced by the reporter cassette [35]. The observed phenotype suggested that Lgr5 exerts a negative regulatory role on the Wnt/β-catenin pathway in stem cells [21]. Following the publication of reports identifying Rspondins as ligands for Lgr5, and showing that this interaction enhances Wnt/β-catenin signaling *in vitro*, we have readdressed the biological relevance of this marker on two other plain knockin/knockout Lgr5eGFP-Ires-CreERT2 and Lgr5-DTReGFP mouse strains, which differ from the Lgr5-LacZNeo one by the fact that reporter cassettes are inserted within the first exon of Lgr5 [1,24]. Irrespective of the mouse strain considered, the observation that Wnt ligand inhibition at the onset of cytodifferentiation could interfere with the precocious Paneth cell differentiation observed at birth in knockouts further strengthened conclusions drawn from the first report on a role of Lgr5 in negative regulation of the Wnt/β-catenin activity during the prenatal stage. Similarly, conditional ablation of the Lgr5 function in the postnatal phase during which Paneth cell differentiation normally proceeds accelerated the maturation process. In adults,

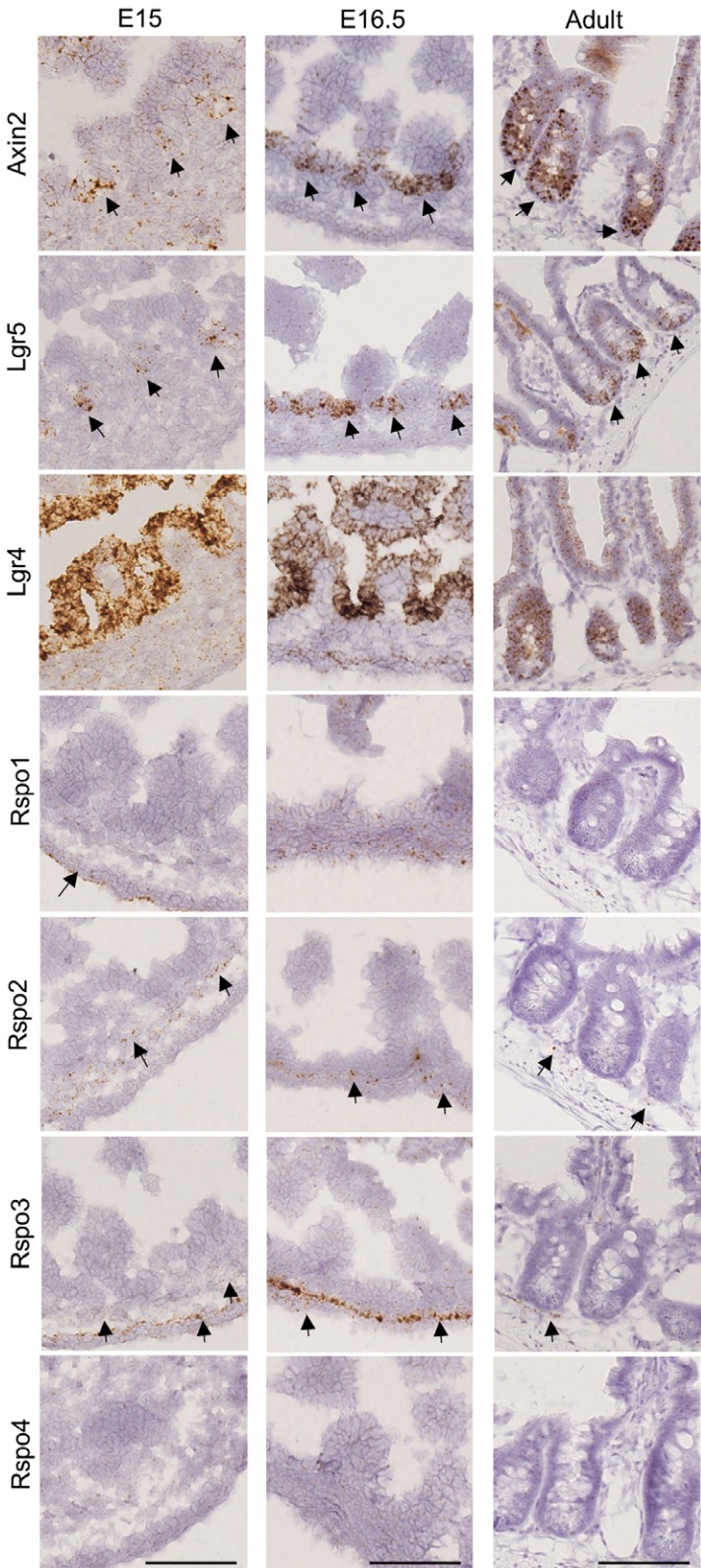

**Figure 5. Expression profile of Rspondin ligands and Lgr receptors during intestinal maturation.**
Expression of Axin2, Lgr5, Lgr4, Rspo1, Rspo2, Rspo3, and Rspo4 genes was analyzed on E15, E16.5, and adult stages by RNAscope. Arrows evidence expressing cells.Data information: Scale bar, 100 μm.

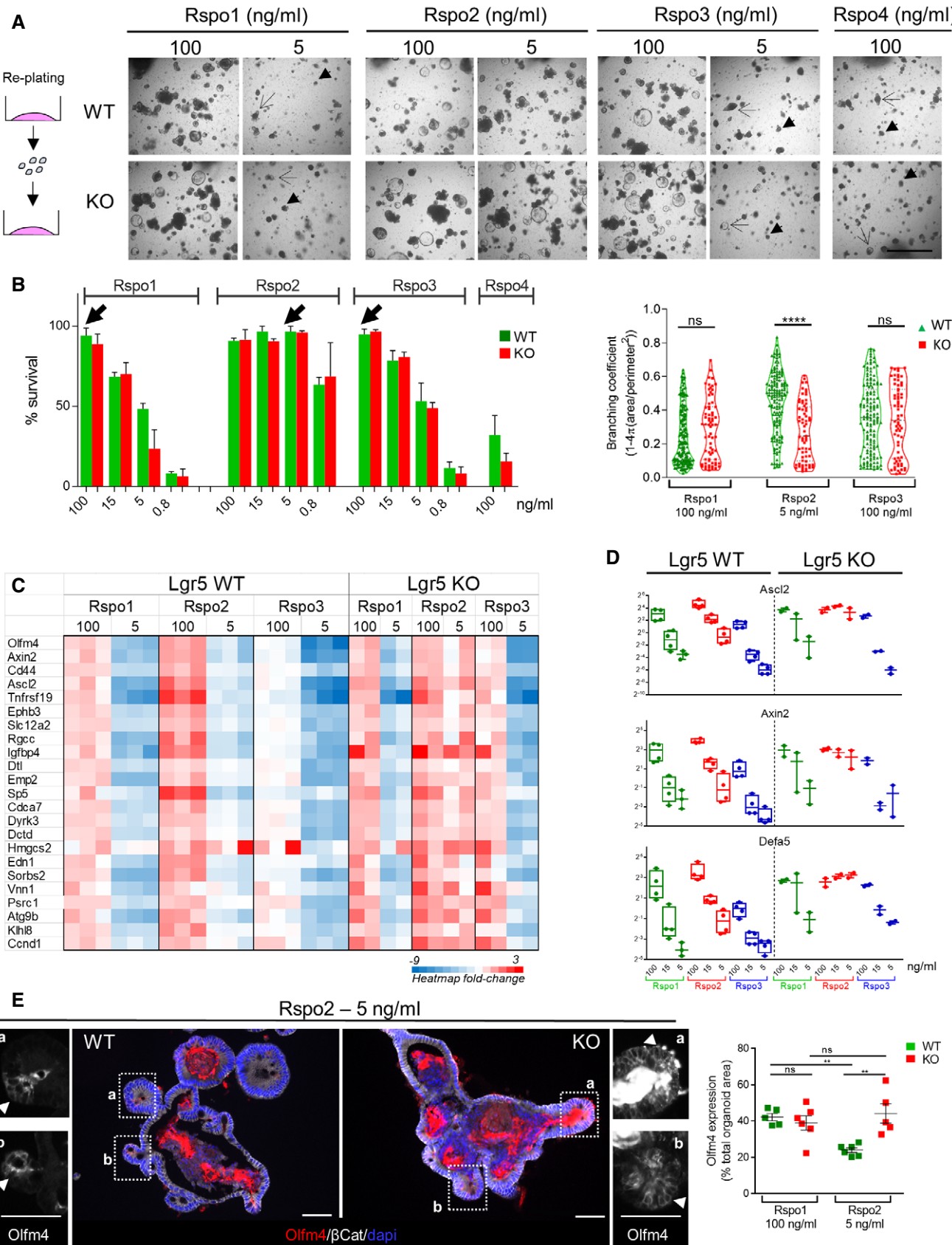

Figure 6.

**Figure 6. Rspondin 2/Lgr5 interaction regulates stem cell fate in organoids.**

A   Representative pictures showing the influence of mouse Rspondin type and concentration on growth of replated Lgr5-DTReGFP WT and Lgr5 KO organoids after 4 days of culture. Arrows show alive elements and arrowheads dead elements.

B   Organoid morphology depends on Rspondin ligand type and concentration. Left panel: Growth quantified as the percentage of elements present at day 1 surviving at day 4. A mean of 100 elements was analyzed per sample and per condition ($n$: 4 WT, 2 KO organoid culture samples, each originating from a given embryo). Right panel: Organoid complexity measured by branching coefficient. Each symbol indicates the value of a given organoid (analysis was done on 4 WT and 2 KO organoid culture samples, each originating from a given embryo).

C   Heatmap showing the impact of Rspondin type and concentration (ng/ml) on relative Wnt target gene expression (list from the Sansom APC dataset) in Lgr5-DTReGFP WT and KO organoids analyzed by RNAseq ($n$ = 3 WT and 2 KO organoid culture samples, each originating from a given embryo).

D   Gene expression analysis by qRT–PCR of stem cell markers in Lgr5-DTREeGFP WT and KO organoids after 5 days of culture (expression relative to reference genes). Each symbol indicates the value for an organoid culture originated from a given embryo.

E   Immunofluorescence showing Olfm4[+ve] cells in Lgr5-DTReGFP WT and KO organoids cultured in Rspo2 conditions. Epithelium is delineated with b-catenin and nuclei counterstained with DAPI (Merge). Insets: White arrowheads evidence Olfm4[+ve] cells concentrated in organoid protrusions. Quantification of Olfm4[+ve] area with respect to the total epithelial area. Each symbol indicates the value for an organoid culture originating from a given embryo.

Data information: Scale bars, 1,000 μm (A) and 50 μm (E). Data are represented as means ± SEM (B left panel, E) and min to max (B right panel and D) ns, not significant; **$P < 0.01$; ****$P < 0.0001$ by Mann–Whitney test (B) and two-way ANOVA followed by Tukey's multiple comparisons test (E).

no overt phenotype was observed upon Lgr5 loss (though significant transcriptomic changes were detected), likely blurred in a context where Paneth cells are fully established.

To investigate the molecular mechanisms associated with the observed phenotype in Lgr5-deficient embryos at birth, the transcriptomes of heterozygous (control) and homozygous (KO) ISCs were compared at E16.5. Lgr5 deficiency induced a profound downregulation of many extracellular matrix (ECM) components, i.e., collagens, proteoglycans (versican, decorin), glycoproteins (laminin a4, fibronectin, thrombospondins, nidogens), and ECM-associated modifying proteins (Adamts proteases, Lysyl oxidase, cytokines, glycosaminoglycan-modifying heparan sulfate sulfotransferase), all involved in matrisome formation and maintenance. The highly dynamic matrisome, defined by a core of ~300 proteins localizing at the basement membrane (between epithelial cells and stromal cells) and in the neighboring interstitial space, is known to regulate tissue development, as well as fibrosis and cancer progression in adults [29]. Softening of the Matrigel in the *ex vivo* culture system has recently been demonstrated to regulate stem cell fate in adult ISC-derived organoids [36]. Coherent with this report, decreased expression of ECM proteins by E16.5 KO ISCs, likely reducing stiffness of the tissue surrounding ISCs, correlated with amplification of the stem cell pool and precocious differentiation of Paneth cells before birth. Moreover, according to our transcriptomic data on adult versus embryonic ISCs, the capacity of stem cells to synthesize their own ECM components progressively decreased during maturation, correlating with acquisition of a definitive "fully" epithelialized phenotype. Nevertheless, adult ISCs still conserved the control on their epithelial status as conditional loss of Lgr5 favored transition from an epithelial to a more mesenchymal-like phenotype (EMT Hallmark upregulated in cKOs Lgr5[Cre/flx] versus HEs Lgr5[Cre/+]). In line with these findings obtained under *in vivo* homeostasis, Lgr5 knockdown on cultured colorectal cancer cell lines also leads to upregulation of EMT-related genes *in vitro* [37]. These results, linking Lgr5 function to EMT control in adult ISCs, highlight the interest to conduct further studies addressing the role of this receptor on the metastatic potential of cancer cells.

In the adult intestinal crypt, the ISC microenvironment substantially differs from the prenatal intervillus domains, in particular through its cellular niche composed of adjacent epithelial Paneth cells as well as various resident stromal cell types, which all might contribute to provide the adequate ECM environment to ISCs

[14,38,39]. Initial *in vitro* studies identified the matricellular proteins Rspondins (Rspo1/Rspo2/Rspo3/Rspo4) as ligands for Lgr4/5/6 paralogues [16–19]. Peri-cryptal intestinal myofibroblasts are reported as the main source of these ligands, predominantly expressing Rspo3 under normal conditions and being able to overexpress Rspo2 upon infection or inflammatory stimuli in adults [40,41]. In the present study, we confirmed these data in normal adult tissues and found a similar prevalence of Rspo2/Rspo3 ligands during development. Then, we compared the effect of the various Rspo ligands on ISC growth and differentiation in the *ex vivo* culture system. First of all, Lgr5-deficient ISCs could grow as organoids with long-term self-renewing capacity, and this implied that Rspondin ligands interact with other receptors in Lgr5-null ISCs. These receptors could possibly be the cognate Lgr4 receptor or likely heparan sulfate proteoglycans, such as glypican or syndecan receptors (namely Gpc3/Gpc4 and Sdc1/Sdc4 that are the most expressed in ISCs Fig EV4C), reported as alternative receptors for Rspo2 and Rspo3 to regulate Wnt/β-catenin signaling [42,43]. Secondly, coherent with reported affinities of Rspondins on cultured HEK293 cells, Rspo2 appeared as the most potent ligand to promote survival, growth, and differentiation of Lgr5 WT or KO organoids [16,42]. Moreover, in agreement with the *in vivo* studies at E16.5, the absence of Lgr5 expression in ISCs *ex vivo* correlated with amplification of the stem cell pool, differentiation toward the Paneth lineage, and altered ECM component production. Such differential behavior as compared to WT organoids was especially noticed when using as ligand Rspo2, the family member showing highest affinity for the Lgr5 receptor [16]. Together, these experiments suggest that the Lgr5 receptor desensitizes ISCs to the Wnt potentiation effect mediated by Rspo2 through its interaction with other receptors. The absence of Lgr5 in ISCs would result in increased Wnt activity and bias toward Paneth cell differentiation (Fig EV5D). These data are in line with a previous report showing that Rspo2, but not other family members, through interaction with the Lgr5 receptor, exhibits tumor suppressive activity on colorectal cancer cell lines via negative regulation of the Wnt/β-catenin pathway [20]. Rspondins are known to be essential to regulate ISC maintenance, playing non-interchangeable functions with Wnt ligands. Considering the complex expression pattern of multiple ligands and receptors, associated with differential affinities and receptor abundances, further studies are needed to decipher how these interactions can coordinately regulate extracellular matrix production in ISCs and control their fate.

# Materials and Methods

### Experimental animals

Animal procedures complied with the guidelines of the European Union and were approved by the local ethics committee (CEBEA from the faculty of Medicine, ULB) under the accepted protocols 535N and 631N. Mice were bred and maintained under a standard 12-h light–dark cycle, with water and rodent chow *ad libitum*. Mice strains were as follows: *Lgr5*-DTR knockin [24], Ctnnb1[exon3] [28], Lgr5-LacZNeo [35], Lgr5-Flox exon16 [17], and Lgr5-GFP-Cre[ERT2] [1], Rosa26CAG floxed stop tdTomato referred as Rosa26R-Tomato and *Axin2*-LacZ (Jax mice). The day the vaginal plug was observed was considered as embryonic day 0.5 (E0.5). For LGK974 rescue experiments, pregnant females were administered with the LGK974 compound (kindly provided by Novartis) by oral gavage once a day at a dose of 1–3 mg/kg/day. For lineage tracing experiments, tamoxifen (Sigma-Aldrich) was dissolved in a sunflower oil (Sigma-Aldrich)/ethanol mixture (19:1) at 10 mg/ml and used at a dose of 0.1 mg/g of body weight for gestating and lactating females or at a single dose of 2 mg for adult mice by intra-peritoneal injection (i. p).

### Tissue processing and immunohistochemical analysis

Small intestine samples were immediately fixed with 10% formalin solution, neutral buffered (Sigma-Aldrich) overnight at + 4°C, and then sedimented through 30% sucrose solution before OCT embedding. Histological and X-gal staining protocols as well as immunofluorescence/histochemistry experiments on 6 μm sections were carried out as previously described [21]. Lendrum's staining was performed according to the manufacturer's instructions (cat # 631340, Clinitech, UK). The primary antibodies used for staining are listed in Table EV1. Samples were visualized with Zeiss Axioplan 2 (immunohistochemistry) or NanoZoomer digital scanner (Hamamatsu) and Zeiss Observer Z1 microscope (immunofluorescence).

In E18.5 tissues, Paneth cells were quantified using Lendrum's staining on a mean of 50 intervillus (IV) regions per sample. In postnatal and adult samples, Paneth cells identified by the Plyz marker were quantified on a mean of 20 and 50 Tomato-recombined crypts, respectively. Postnatal and adult lineage tracing analysis was performed on a mean of 100 crypt-villus units. Quantification of Olfm4[+ve] cells in E18.5 tissues was performed on a minimum of 20 IV per sample. The number of animals used for each experiment is reported in Figures or in Figure legends.

### Ex vivo culture

Embryonic small intestine was dissociated with 5 mM EDTA-in DPBS (Gibco) according to the protocol reported in [Ref. 44]. Briefly, the culture medium used consisted in Advanced-DMEM/F12 medium supplemented with 2 mM L-glutamine, N2 and B27 w/o vit.A, gentamycin, penicillin-streptomycin cocktail, 10 mM HEPES (all from Invitrogen), 1 mM N acetyl cysteine (Sigma), 50 ng/ml EGF, and 100 ng/ml noggin (both from PeproTech). CHO-derived mouse Rspondins (R&D System) were used at the final concentrations indicated in text and figures. The final concentration of Rspondins in the culture medium was initially tested in pilot experiments in order to add comparable amounts of bioactive ligands (bioactivity

measured by TOPflash assays and organoid growth curves, R&D systems). Culture medium was changed each other day, and after 5–6 days in culture, organoids were harvested, mechanically dissociated, and replated in fresh Matrigel (BD Biosciences). Culture media were supplemented with 10 μM Y-27632 (Sigma-Aldrich) in all initial seeding and replating experiments. Pictures were acquired with a Moticam Pro camera connected to Motic AE31 microscope. Branching coefficient for each individual organoid was calculated with the formula :[1-4π(area/perimeter$^2$)] as previously described [45]. Area and perimeter of a mean of 30 organoids (between 13 and 42 organoids) were calculated using the ImageJ software per condition for 4 WT and 2 KO independent cultures.

### Flow cytometric analysis and cell sorting (FACS)

Embryonic small intestine or adult intestinal crypts (isolated with 5 mM EDTA) were dissociated with the TrypLE cell dissociation reagent (Thermo Fisher) and passed through a 40-μm nylon cell strainer (Greiner). For sorting experiments involving the Lgr5[Cre]-RosaTomato lines, tamoxifen was administered by i.p. 48 h before tissue harvesting. For RNAseq analysis, after cell gating and doublets exclusion, ISCs EGFP[+ve] cells were directly sorted using the FITC channel (Lgr5-DTReGFP line) or were selected as recombined EGFP[+ve]/Tomato[+ve] cells using the FITC and PE channels (Lgr5[Cre]-RosaTomato line). Sorted cells were collected over QIAzol lysis reagent (Qiagen).

### Gene expression analysis

qRT–PCR was performed on total RNA extracted from embryonic tissues or organoid cultures as reported [21]. Expression levels were normalized to that of reference genes (Rpl13, Gapdh, Ywhaz). Primer sequences are reported in Table EV1. *In situ* hybridization experiments were performed according to manufacturer instructions with the RNAscope kit (ACD-Biotechne) (probes listed in Table EV1).

### RNA seq and Gene Set Enrichment Analysis (GSEA)

RNA quality was checked using a Bioanalyzer 2100 (Agilent technologies). Indexed cDNA libraries were obtained using the Ovation SoLo (NuGen) or the NEBNext RNA-Seq Systems following manufacturer recommendation. The multiplexed libraries were loaded on a NovaSeq 6000 (Illumina) using a S2 flow cell, and sequences were produced using a 200 Cycle Kit. Paired-end reads were mapped against the mouse reference genome GRCm38 using STAR software to generate read alignments for each sample. Annotations Mus_-musculus.GRCm38.90.gtf were obtained from ftp.Ensembl.org. After transcripts assembling, gene-level counts were obtained using HTSeq. Genes differentially expressed were identified with EdgeR method and further analyzed using GSEA MolSig (Broad Institute) [46]. Gene pattern was used to compare by pre-ranked GSEA E16.5 Lgr5 KO versus HE; Adult HE versus E18.5 HE, or Adult Lgr5 KO versus HE transcriptomes with gene datasets [47]. Gene Ontology was used to identify biological processes [48].

### Statistical analysis

Statistical analyses were performed with GraphPad Prism 5. All experimental data are expressed as mean ± SEM. The significance

of differences between groups was determined by appropriate parametric or non-parametric tests as described in Figure legends.

## Data availability

The datasets produced in this study are available in the following databases: RNAseq data: Gene Expression Omnibus GSE135362 (https://www.ncbi.nlm.nih.gov/geo/query/acc.cgi?acc = GSE135362)—Datasets EV1–EV4.

**Expanded View** for this article is available online.

### Acknowledgements

We are grateful to Genentech for providing us with the *Lgr5*-DTReGFP mice, Novartis for providing LGK974 compound, Makoto Taketo for providing the Ctnnb1 exon3 strain, and Hans Clevers for providing the Lgr5-LacZNeo and Lgr5 flox mouse strains. Interuniversity Attraction Poles Programme-Belgian State-Belgian Science Policy (6/14), Fonds De La Recherche Scientifique FNRS (CDR/OL J.0083.20, Télévie 7459317F, Fonds pour la Formation à la Recherche dans l'Industrie et dans l'Agriculture FNRS 1.E.108.20F), Fonds de la Recherche Scientifique Médicale of Belgium, Walloon Region (program CIBLES), and non-for-profit Association Recherche Biomédicale et Diagnostic.

### Author contributions

VFV: study concept and design, acquisition of data, analysis and interpretation of data, statistical analysis, and drafting of the ms. ML, RG, DR-S, AL, and FL: acquisition of data, analysis and interpretation of data, and statistical analysis. GV: study concept and design, critical revision of the ms, obtained funding, and study supervision. M-IG: study concept and design, acquisition of data, analysis and interpretation of data, drafting of the ms, critical revision of the ms, and study supervision.

### Conflict of interest

The authors declare that they have no conflict of interest.

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
