## [Review Process File · EMBO Reports]

Lgr5 Controls Extracellular Matrix Production By Stem Cells In The Developing Intestine

Valeria Fernandez-Vallone, Morgane Leprovots, Didac Ribatallada-Soriano, Romain Gerbier, Anne Lefort, Frédéric Libert, Gilbert Vassart, and Marie-Isabelle Garcia

DOI: [10.15252/embr.201949224](https://doi.org/10.15252/embr.201949224)

Corresponding author(s): Marie-Isabelle Garcia (mgarcia@ulb.ac.be)

Review Timeline:

Submission Date:	5th Sep 19
Editorial Decision:	8th Oct 19
Revision Received:	24th Feb 20
Editorial Decision:	23rd Mar 20
Revision Received:	14th Apr 20
Editorial Decision:	28th Apr 20
Revision Received:	30th Apr 20
Accepted:	30th Apr 20

Editor: Achim Breiling

Transaction Report:

Dear Dr. Garcia,

Thank you for the submission of your research manuscript to EMBO reports. We have now received reports from the three referees that were asked to assess the manuscript, which can be found at the end of this email.

As you will see, all referees think that the findings are of high interest, but they also have several comments, concerns and suggestions, indicating that a major revision of the manuscript is necessary to allow publication in EMBO reports. As the reports are below, and I think all points need to be addressed, I will not detail them here.

Given the constructive referee comments, we would like to invite you to revise your manuscript with the understanding that all referee concerns must be addressed in the revised manuscript and in a detailed point-by-point response. Acceptance of your manuscript will depend on a positive outcome of a second round of review. It is EMBO reports policy to allow a single round of revision only and acceptance or rejection of the manuscript will therefore depend on the completeness of your responses included in the next, final version of the manuscript.

Revised manuscripts should be submitted within three months of a request for revision; they will otherwise be treated as new submissions. Please contact me if a 3-months time frame is not sufficient so that we can discuss the revisions further.

When submitting your revised manuscript, please also carefully review the instructions that follow below. Failure to include requested items will delay the evaluation of your revision. When submitting your revised manuscript, we will require:

1) a .docx formatted version of the final manuscript text (including legends for main figures, EV figures and tables), but without the figures included. Please make sure that the changes are highlighted to be clearly visible. Figure legends should be compiled at the end of the manuscript text.

2) individual production quality figure files as .eps, .tif, .jpg (one file per figure), of main figures and EV figures. Please upload these as separate, individual files upon re-submission.

The Expanded View format, which will be displayed in the main HTML of the paper in a collapsible format, has replaced the Supplementary information. You can submit up to 5 images as Expanded View. Please follow the nomenclature Figure EV1, Figure EV2 etc. The figure legend for these should be included in the main manuscript document file in a section called Expanded View Figure Legends after the main Figure Legends section. Additional Supplementary material should be supplied as a single pdf labeled Appendix. The Appendix should have page numbers and needs to include a table of content on the first page (with page numbers) and legends for all content. Please follow the nomenclature Appendix Figure Sx, Appendix Table Sx etc. throughout the text, and also label the figures and tables according to this nomenclature.

For more details please refer to our guide to authors:

See also our guide for figure preparation:

4) a complete author checklist, which you can download from our author guidelines (<https://www.embopress.org/page/journal/14693178/authorguide>). Please insert page numbers in the checklist to indicate where the requested information can be found in the manuscript. The completed author checklist will also be part of the RPF.

Please also follow our guidelines for the use of living organisms, and the respective reporting guidelines: <http://www.embopress.org/page/journal/14693178/authorguide#livingorganisms>

5) that primary datasets produced in this study (e.g. RNA-seq. data) are deposited in an appropriate public database. See: <http://embor.embopress.org/authorguide#datadeposition>

The accession numbers and database should be listed in a formal "Data Availability " section (placed after Materials & Methods) that follows the model below. Please note that the Data Availability Section is restricted to new primary data that are part of this study.

Data availability

6) We strongly encourage the publication of original source data with the aim of making primary data more accessible and transparent to the reader. The source data will be published in a separate source data file online along with the accepted manuscript and will be linked to the relevant figure. If you would like to use this opportunity, please submit the source data (for example scans of entire gels or blots, data points of graphs in an excel sheet, additional images, etc.) of your key experiments together with the revised manuscript. If you want to provide source data, please include size markers for scans of entire gels, label the scans with figure and panel number, and send one PDF file per figure.

7) Our journal encourages inclusion of *data citations in the reference list* to directly cite datasets that were re-used and obtained from public databases. Data citations in the article text are distinct from normal bibliographical citations and should directly link to the database records from which the

data can be accessed. In the main text, data citations are formatted as follows: "Data ref: Smith et al, 2001" or "Data ref: NCBI Sequence Read Archive PRJNA342805, 2017". In the Reference list, data citations must be labeled with "[DATASET]". A data reference must provide the database name, accession number/identifiers and a resolvable link to the landing page from which the data can be accessed at the end of the reference. Further instructions are available at: <http://www.embopress.org/page/journal/14693178/authorguide#referencesformat>

8) Regarding data quantification and statistics, can you please specify, where applicable, the number "n" for how many independent experiments (biological replicates) were performed, the bars and error bars (e.g. SEM, SD) and the test used to calculate p-values in the respective figure legends. Please provide statistical testing where applicable, and also add a paragraph detailing this to the methods section. See: <http://www.embopress.org/page/journal/14693178/authorguide#statisticalanalysis>

I look forward to seeing a revised version of your manuscript when it is ready. Please let me know if you have questions or comments regarding the revision.

Yours sincerely,

Achim Breiling
Editor
EMBO Reports

Referee #1:

In this manuscript, Vallone and colleagues explore the role of Lgr5 receptors, their ligands - the Rspodins - and associated Wnt signaling in intestinal stem cell maturation using Lgr5-KO mice. They report that loss of Lgr5 receptors induces overactivation of the Wnt/beta-catenin pathway in intestinal epithelium, which promotes early cytodifferentiation of Paneth cell lineages during development. The precociously mature epithelial phenotype of intestinal stem cells in Lgr5-KO intestine was supported by RNA sequencing and gene enrichment analysis. They also report differential effects of the Rspodin family (Rspodin 1 and Rspodin 2) through Lgr5 on intestinal epithelial organoids.

The results include interesting data and the findings can be potentially useful for understanding the mechanism by which Rspodin works through Lgr receptors on Wnt pathways. However, the manuscript was presented in a way that there appeared to be three related but independent results and experiments (Figures 1-4, Figure 5 and Figure 6). The connection between the components was weak and it does not feel like a cohesive story. A few concerns need to be addressed:

1. The authors titled the manuscript "Lgr5 controls extracellular matrix production by stem cells in the developing intestine". However, the ECM was only brought up in Figure 4 and the point did not feel like the main takeaway from this manuscript. The correlation between Lgr5-KO and decreased EMT marker genes (including ECM) may be suggestive of a mature phenotype of ISCs, but the authors did not conduct further experiments to show a role for the extracellular matrix. This merits further investigation or should be removed.

2. In figure 1B, it is striking to see that despite the significant upregulation of some Paneth cell

markers such as Crypt4 and Crypt5, other classical Paneth cell markers (such as Lyz1) were not as upregulated. Do these Crypt4/5-high epithelial cells really represent differentiated Paneth cells? The authors should provide some explanation or discussion as to why only some of these genes are upregulated and what they think might be going on.

3. In figure 2, the authors show a reduced level of Wnt related genes and Paneth cell differentiation with Lgr5-KO intestine treated with LGK974. The authors should also provide some images of the adult and quantification of Olfm stained tissue section to identify whether Wnt downregulation is due to the reduced stem cell number or reduced Wnt signaling per stem cells.

4. The b-cat staining in figure 2 is not interpretable and should either be improved or removed.

5. In figure 3B&D, the Plyz stained images shown are too zoomed out and it is unclear how the number of cells were quantified. The adult Lgr5(Cre/fl) seem to have the lowest level of Plyz staining, but the quantification shows otherwise. The authors should counter stain with a stem-cell marker, to make clear how the number of Plyz+ cells were quantified. Further support of the data with other methods such as flow cytometry (e.g. how many Paneth cells among RFP+, Epcam+, Cd44+ population) should be provided.

6. In figure 6C, it is unclear what the phenotypes are that readers should be observing from the data. In addition, reduction in the proportion of Olfm4-expressing cells in WT organoids in Rspo2 vs Rspo1 condition is not obvious from the images shown. The authors should indicate the phenotype observed with arrowheads and make appropriate quantification.

7. The differential organoid phenotypes between Rspodin1 and Rspodin2 observed in WT organoids were not observed in Lgr5-KO organoids (figure 6). The authors did not elaborate on the observation other than to state that "these data indicated that Rspo ligands do not exhibit redundant functions on ISCs" (Line 216-217). The observation is interesting and should be discussed in more depth.

Minor comment:

1. Line 42-44: the sentence is hard to understand.

2. The official gene names for Crypt5 and Crypt4 should be provided.

3. Figure 2B does not add as much information as Figure S2D, so consider switching figure S2D and Figure 2B.

4. For RT-qPCR(Fig 3) and transcriptomic analysis(Fig 4), the authors mention analyzing Tom-recombined cells and GFP+ ISC precursor cells, respectively. Details on sample preparation (cell sorting) should be described in the Methods.

Referee #2:

This manuscript by Vallone et al. builds on their previous work to conclusively demonstrate that loss of Lgr5 in the fetal gut leads to enhanced Paneth cell differentiation and stem cell marker expression. This phenotype was reversed by systematically inhibiting Wnt secretion, thus demonstrating that Lgr5 suppresses Wnt signaling specifically during fetal development. The authors also transcriptionally profiled Lgr5GFP+ cells and found that loss of Lgr5 resulted in reduced extracellular matrix production and increased maturation of ISCs. Lastly, the authors demonstrate that Rspo2 has non-redundant effects on the growth of intestinal organoids.

This is an interesting study that attempts to provide mechanistic insight into the differential role of Lgr5 in the fetal vs the adult gut. Given the broad interest for Lgr5 as a stem cell marker in various tissues, this study should be of interest to a wide readership. Although slightly incremental in nature, the first three figures are well documented and confirm the importance of Lgr5 in regulating Paneth cell/ISC differentiation. The novel idea brought forward by this manuscript is that Rspo2 and Lgr5 may act together to suppress Wnt signaling during development. Unfortunately, this aspect of the study was not investigated in sufficient detail. Through more refined expression studies, the authors could have examined the relative expression of Rspo2 and Lgr5 in the embryo and postnatally, as well as further investigated the effects of Rspo2 on organoids. Therefore, the following points should be addressed before publication in EMBO reports.

Specific comments:

Several basic questions are left unanswered by this study regarding the expression of Rspo and Lgr5:

Could changes in Rspo2 expression between the embryo and adult underlie differential role of Lgr5 during the stages. Is Rspo2 (and Rspo 1/3) differentially expressed in the fetal vs adult intestine? Are Rspo2 expressing cells present near Lgr5 clusters in the fetal intestine?

In Figure 6 the authors show that treatment with Rspo2 induces a long crypt phenotype. The expression studies also indicated that Rspo2 is less potent than Rspo1 in activating ISC markers. Thus, are the differences in crypt morphology a reflection of lower Wnt signaling activity in Rspo2 treated organoids. Does treatment of WT organoids with lower doses of Rspo1 induce long protrusions as well? In other words, are these effects due to the lowered potency of Rspo2 to activate Wnt signaling or due to Wnt-independent Rspo2/Lgr5 functions on crypt morphology. It would have been very informative to perform RNAseq analysis on the Rspo2 vs Rspo1 treated organoids to help discriminate between these two possibilities.

Based on Fig4D, the authors conclude that a conversion of autocrine Wnt ligand expression occurs from Wnt 5a in fetal intestines to Wnt 3 in adults. This is a surprising result given that previous reports have shown that Wnt5a is a mesenchymal factor and Wnt3 is expressed in Paneth cells. Why do the authors conclude that these are autocrine signals? The conclusions drawn from the transcriptional profiling of Lgr5GFP+ cells should be validated by ISH.

Referee #3:

"Lgr5 Controls Extracellular Matrix Production By Stem Cells In the Developing Intestine", submitted by Vallone, V. et al, examines the functional consequence of inactivating Lgr5 on the development of the intestine. The authors find that inactivation of Lgr5 during development results in precocious Paneth cell differentiation, an increase in WNT tone, and expanded stem cell compartment. They further show that these phenotypes are dependent on WNT, as experimentally reducing Wnt signaling eliminates the difference between knock outs and controls. The authors further explore the functional consequences of inactivating Lgr5 using RNAseq and identify a difference in the in vitro response of two difference LGR5 ligands, Rspo1 and Rspo2.

Overall the paper is well written and easy to follow. The experiments are well performed and the findings of general interest. Although the Paneth and stem cell phenotype has previously been

described, the authors provide additional mechanistic insight and demonstrate the dependence of the phenotype on WNT. The paper then shifts gears and reports on an RNAseq experiment and in vitro response to RSPOs. The findings of these experiments are quite interesting and novel, but require additional follow up to assess their relevance. As written, the manuscript does not have a single key finding; there are several interesting observations that are not demonstrated to be connected. I believe this can be improved with additional experiments.

MAJOR:

The power of the RNAseq experiment is to determine the differences between control and LGR5 KO stem cells in embryos, in adults, over developmental time, and whether there is an interaction between genotype and developmental time. The manuscript would be improved if these data were presented more concisely. For example, two graphs (4B and 4E) are presented with some redundant data between the graphs. The authors could use one graph that would include differences between 16.5 HEvKO, Adult HEvKO, 16.5HE v Adult HE. This would make it easier to visualize the interaction between genotype and time that the authors suggest. Furthermore, the authors should explore statistical methods to use to assess the significance of this interaction.

The manuscript will be further improved by following up on what was identified in the RNAseq experiment. The authors should include PCR, and tissue-based techniques, exploring the effects on ECM production and inflammatory response. This should be done with a developmental sequence in control and KOs. Furthermore, it would really tie the manuscript together to include embryos from LGK974 treated dams. This will help assess whether the same processes that lead to precocious Paneth cell differentiation also affects ECM production and inflammatory responses.

The second novel observation regarding the differential effect of RSPO1 and RSPO2 in culture needs to be further explored. RSPO2 is much more potent than RSPO1 and has a dose-response curve that is an inverted U. Depending on what part of the curve one is assessing might lead to disparate conclusions. The authors need to show a dose-response of both RSPO1 and RSPO2 using concentrations lower than 15 ng/ml. For example, a curve of .1, 1, 10, 50, 100 might be a good place to start.

MINOR:

1. The authors should provide a close up of the intervillus region in Figure 2B. It is very difficult to see any effects on beta-Catenin in the low power image.

2. How are Paneth cells defined in Figure 2C? Can the authors include PCR data on additional Paneth cell markers, not just Crypt5? It would be nice to include the same markers that were used to describe the phenotype in Figure 1.

3. It is unclear why two different periods of chase were used in early PN versus adult animals in Figure 3. The authors should be consistent and use a 10 day chase in the adults. Longer chases may be confounded by differences in Paneth cell lifespan.

4. It may be possible to make the manuscript more concise by combining Figures 1 and 5. In addition, Figure 4D can be moved to Extended Data and this figure can be simplified by combining graphs B and E.

Referee #1:

In this manuscript, Vallone and colleagues explore the role of Lgr5 receptors, their ligands - the Rspodins - and associated Wnt signaling in intestinal stem cell maturation using Lgr5-KO mice. They report that loss of Lgr5 receptors induces overactivation of the Wnt/beta-catenin pathway in intestinal epithelium, which promotes early cytodifferentiation of Paneth cell lineages during development. The precociously mature epithelial phenotype of intestinal stem cells in Lgr5-KO intestine was supported by RNA sequencing and gene enrichment analysis. They also report differential effects of the Rspodin family (Rspodin 1 and Rspodin 2) through Lgr5 on intestinal epithelial organoids.

*The results include interesting data and the findings can be potentially useful for understanding the mechanism by which Rspodin works through Lgr receptors on Wnt pathways. However, the manuscript was presented in a way that there appeared to be three related but independent results and experiments (Figures 1-4, Figure 5 and Figure 6). The **connection between the components was weak and it does not feel like a cohesive story**. A few concerns need to be addressed:*

The revised manuscript (ms) has integrated new data generated from RNaseq performed on organoid cultures showing that Lgr5 can regulate Wnt signaling and ECM components synthesis via the Rspo2 ligand (new Fig. 6). In addition, we have combined previous Fig. 1 and Fig. 5 into a new Fig. 1. The expression pattern of Rspodin ligands and Lgr4/Lgr5 receptors during intestinal development and in adult tissues is now provided as a new Fig. 5. The text and discussion have been completely revised accordingly, all this improving cohesiveness to the whole manuscript.

*1. The authors titled the manuscript "Lgr5 controls extracellular matrix production by stem cells in the developing intestine". However, the ECM was only brought up in Figure 4 and the point did not feel like the main takeaway from this manuscript. The correlation between Lgr5-KO and decreased EMT marker genes (including ECM) may be suggestive of a mature phenotype of ISCs, but the authors did not **conduct further experiments to show a role for the extracellular matrix**. This merits further investigation or should be removed.*

The finding that fetal Lgr5 ISCs express significant levels of ECM markers at early stages and that Lgr5 deficiency affects stem cell production of these molecules was brought only late in the ms simply because this was only detected by RNaseq analysis. Since we observed that ISC maturation (adult versus embryo) is also associated with reduction in the ECM content, it is likely that the observed phenotype for Lgr5 KO embryos is related to premature maturation of these ISCs. In the revised ms, we present new experiments performed on organoid cultures using the various Rspodin ligands. We provide evidences (RNaseq data on organoid cultures) that the Lgr5/Rspo2 interaction regulates Wnt signaling activity and impacts on ECM component synthesis; which further sustains the choice of the

title of this ms. For sure, additional studies investigating molecular mechanisms linking Lgr5/Wnt signaling/ECM, will constitute a major point to be addressed in future.

2. In figure 1B, it is striking to see that despite the significant upregulation of some Paneth cell markers such as *Crypt4* and *Crypt5*, other classical Paneth cell markers (such as *Lyz1*) were not as upregulated. Do these *Crypt4/5*-high epithelial cells really represent differentiated Paneth cells? The authors **should provide some explanation** or discussion as to why only some of these genes are upregulated and what they think might be going on.

Paneth cell maturation normally occurs coincident with crypt formation after birth in the mouse. The Paneth cell differentiation program involves sequential expression of defensins, metalloprotease 7 and P-Lysozyme, which expression becomes detected later during the maturation process (around postnatal day 6-10) (Bry et al. 1994; Inoue et al. 2008)*. Therefore, in our study performed on E18.5 embryos, precocious maturation for this lineage is quantified using the well-recognized Lendrum's staining method detecting defensins with Phloxine-tartrazine dyes.

In the revised ms, we have increased the n of individuals for the 3 genotypes (now n=8 WT, 11 HE and 10 KO) to study paneth cell markers' expression by quantitative RT-PCR. These additional samples confirm premature differentiation of Lgr5-deficient embryos with *Defa5*, *Defa4*, *Mmp7* and *Lyz1* being detected upregulated as compared to wild-type littermates. Figure 1 and its legend have been modified accordingly.

*Bry et al. Doi 10.1073/pnas.91.22.10335

Inoue et al. <https://doi.org/10.1111/j.1574-695X.2008.00390.x>

3. In figure 2, the authors show a reduced level of Wnt related genes and Paneth cell differentiation with Lgr5-KO intestine treated with LGK974. The authors should also provide **some images of the adult and quantification of Olfm stained tissue section** to identify whether Wnt downregulation is due to the reduced stem cell number or reduced Wnt signaling per stem cells.

As suggested, we have now added quantification of Olfm4-positive cells per intervillus region in untreated and LGK974-treated tissues in wild-type and knockout embryos. These data indicate that Wnt inhibition drastically reduces stem cell number, especially in the ileum. To avoid redundancy, the former Fig. 1C (showing Olfm4-positive quantification in control tissues) is now integrated to the new Fig. 2C. Unfortunately, although initially collected, tissues from treated adult dams have been lost (due to -80°C freezer failure) and additional experiments were out of time for this revision. However, Liu et al. (2013) have previously reported that at a similar dose, LGK974 does not significantly affect adult tissues.

Note that to avoid redundancy in Figures, stem cell pool amplification in E16.5 Lgr5 KO embryos (formerly measured by Olfm4 counting in Fig. 1C) is now integrated in the new Fig. 2C. Amplification of the ISC pool has been replaced by a new Fig. 1D showing eGFP⁺ve ISCs pool amplification analyzed by FACS.

4. The b-cat staining in figure 2 is **not interpretable** and should either be improved or removed.

The former Fig. 2B has been replaced by a new Fig. 2B. The new panels show immunofluorescence for b-catenin/Olfm4/DAPI in control and treated tissues (See also above the answer to the point 3).

5. In figure 3B&D, the Plyz stained images shown are **too zoomed out** and it is unclear how the number of cells were quantified. The adult Lgr5(Cre/fl) seem to have the lowest level of Plyz staining, but the quantification shows otherwise. The authors should counter stain with a stem-cell marker, to **make clear how the number of Plyz+ cells were quantified**. Further support of the data with other methods such as flow cytometry (e.g. how many Paneth cells among RFP+, Epcam+, Cd44+

population) should be provided.

We agree that the lower intensity of the Plyz staining on the image chosen for Lgr5Cre/Flox tissues might have led to misinterpretation. We are now providing new images, some of them obtained from new experiments (integrating Referee 3, minor point 3) showing co-stainings for Plyz/RFP/DAPI (new Fig. 3C) and Plyz/b-catenin (Fig. EV3B). Moreover, we have clarified in Materials and Methods the way quantification of Plyz-positive cells was done on Tomato (RFP-positive)-recombined crypts. In our opinion, the additional suggested methods might not bring better specificity in the quantification of Paneth cells.

6. In figure 6C, it is unclear **what the phenotypes are that readers should be observing from the data**. In addition, reduction in the proportion of Olfm4-expressing cells in WT organoids in Rspo2 vs Rspo1 condition is not obvious from the images shown. The authors should indicate the phenotype observed with arrowheads and make appropriate quantification.

The Olfm4 protein is secreted in the surrounding medium and in the context of organoids, it remains in the central lumen of organoids, complicating the visualization of the positive cells. We provide in the new Fig. 6E additional insets of the organoid protrusions (focusing on WT and KO organoids from Rspo2 cultures) to better evidence the Olfm4+ve cells. Moreover, as suggested, quantification of Olfm4-positive cells in organoids is now provided in the new Fig. 6E. Immunofluorescence stainings confirm our qRT-PCR and RNAseq data, suggesting that Lgr5/Rspo2 regulates the number of ISCs on organoids.

7. The differential organoid phenotypes between Rspodin1 and Rspodin2 observed in WT organoids were not observed in Lgr5-KO organoids (figure 6). The authors did not elaborate on the observation other than to state that "these data indicated that Rspo ligands do not exhibit redundant functions on ISCs" (Line 216-217). The observation is interesting and **should be discussed in more depth**.

We have performed additional experiments to compare the effect of the 4 Rspodin ligands at several concentrations on organoid growth and morphology as well as on gene expression. These data are presented in the new Fig. 6. See also answer to referee 2 specific comment point 2 and referee 3 major point 3. The Discussion section has been revised to further comment on this phenotype.

Minor comment:

1. Line 42-44: the sentence is hard to understand.

The sentence has been replaced.

2. **The official gene names for Crypt5 and Crypt4 should be provided.**

The official gene names for defensins (Defa5, Defa4) have been corrected in the new Fig. 1B, Fig. 2E, Fig. 6D as well as on Fig. EV2C, and Fig. EV2D.

3. Figure 2B does not add as much information as Figure S2D, so consider switching figure S2D and Figure 2B.

Experiments presented in figure S2D were meant to determine whether upregulation of the Wnt/b-catenin pathway in precursor ISCs (Lgr5-positive) could rescue the observed premature Paneth cell differentiation phenotype in Lgr5-deficient embryos. This is the opposite strategy of the experiments presented in the former main Fig. 2B. As explained in the text, reduction of the premature differentiation was achieved only by reducing the Wnt tone with LGK974 treatment within a narrow window of time during gestation. The sentence describing these experiments has been modified in the text to avoid potential misunderstanding.

4. For RT-qPCR(Fig 3) and transcriptomic analysis(Fig 4), the authors mention analyzing Tom-recombined cells and GFP+ ISC precursor cells, respectively. **Details on sample preparation (cell**

sorting) should be described in the Methods.

We apologize for having omitted the FACS sorting procedure used for isolation of Lgr5+ve ISCs in the previous version of the ms. We have now added a paragraph in Materials and methods entitled "Flow cytometric analysis and cell sorting (FACS)".

Referee #2:

This manuscript by Vallone et al. builds on their previous work to conclusively demonstrate that loss of Lgr5 in the fetal gut leads to enhanced Paneth cell differentiation and stem cell marker expression. This phenotype was reversed by systematically inhibiting Wnt secretion, thus demonstrating that Lgr5 suppresses Wnt signaling specifically during fetal development. The authors also transcriptionally profiled Lgr5GFP+ cells and found that loss of Lgr5 resulted in reduced extracellular matrix production and increased maturation of ISCs. Lastly, the authors demonstrate that Rspo2 has non-redundant effects on the growth of intestinal organoids.

*This is an interesting study that attempts to provide mechanistic insight into the differential role of Lgr5 in the fetal vs the adult gut. Given the broad interest for Lgr5 as a stem cell marker in various tissues, this study should be of interest to a wide readership. Although slightly incremental in nature, the first three figures are well documented and confirm the importance of Lgr5 in regulating Paneth cell/ISC differentiation. The novel idea brought forward by this manuscript is that Rspo2 and Lgr5 may act together to suppress Wnt signaling during development. Unfortunately, *this aspect of the study was not investigated in sufficient detail*. Through more refined expression studies, the authors could have examined the relative *expression of Rspo2 and Lgr5 in the embryo and postnatally, as well as further investigated the effects of Rspo2 on organoids*. Therefore, the following points should be addressed before publication in EMBO reports.*

Additional experiments have been performed to address these points (see specific answers below).

Specific comments:

1. Several basic questions are left unanswered by this study regarding the expression of Rspo and Lgr5:

Could changes in Rspo2 expression between the embryo and adult underlie differential role of Lgr5 during the stages. Is Rspo2 (and Rspo 1/3) differentially expressed in the fetal vs adult intestine? Are Rspo2 expressing cells present near Lgr5 clusters in the fetal intestine?

To answer these pertinent questions, we provide in the revised ms the complete expression profile of the 4 Rspodin ligands during development (just after the onset of villus formation-E15 and a later prenatal stage-E16.5) and in adult tissues in the new Fig. 5. Firstly, expression profiles substantially differ for these ligands. Second, Rspo2 and Rspo3 are the ligands showing the highest expression levels in tissues, in stromal cells close to Lgr5 ISCs. These findings are now integrated to the new Figure 5, presented in text and discussed.

2. *In Figure 6 the authors show that treatment with Rspo2 induces a long crypt phenotype. The expression studies also indicated that Rspo2 is less potent than Rspo1 in activating ISC markers. Thus, are the differences in crypt morphology a reflection of lower Wnt signaling activity in Rspo2 treated organoids. Does treatment of WT organoids with lower doses of Rspo1 induce long protrusions as well? In other words, are these effects due to the lowered potency of Rspo2 to activate Wnt signaling or due to Wnt-independent Rspo2/Lgr5 functions on crypt morphology. It would have been very informative to perform RNAseq analysis on the Rspo2 vs Rspo1 treated organoids to help discriminate between these two possibilities.*

First of all, as indicated in lines 339-342 of the former version, the rationale for using defined Rspodin concentrations was based on both i) the reported bioactivity measured in TOPflash assays (reported by the manufacturer) and ii) pilot experiments on organoids to evaluate organoid growth (pilots

experiments not shown in the former version). Both informations suggested higher potency of Rspo2 versus Rspo1, this explaining why we chose to use 100 ng/ml Rspo1 and 5 ng/ml Rspo2.

To determine if morphological differences observed on organoids are related to modulation of Wnt levels, we have performed new culture experiments using the 4 Rspodins at various concentrations (100, 15, 5, 0.8 ng/ml), as also suggested by Referee 3 (see below) and presented these new data in a new Fig. 6. In these new experiments, we have now quantified both stem cell survival (represented by the proportion of living elements at day 4 versus day 1 in matched-fields-new Fig. 6B left panel) and organoid morphology (new Fig. 6B right panel). For this latter parameter, measurement of crypt elongation has been replaced by the more representative analysis of the branching coefficient that integrates area and perimeter of the organoids. The findings are the followings:

- i) Considering ligand bioactivity as the capacity to sustain efficient organoid growth, Rspo4 appeared much less active as compared to the 3 other members (only 30% organoid survival at the highest concentration tested (100 ng/ml) irrespective of the genotype. Reduction of Rspo1 and Rspo3 concentrations from 100 to 15 and 5 ng/ml significantly decreased organoid survival in a dose-dependent manner. At a concentration of 0.8 ng/ml, organoid growth was only still observed on Rspo2 wells. In addition, the viability in presence of Rspo2 showed a “plateau” at any concentration between 100 and 5 ng/ml irrespective of the genotype, suggestive of a higher activity of this ligand in these culture conditions (new Fig. 6A and B). Morphological analysis of organoids showed that protrusion elongation was not especially associated with reduced Rspo concentration (new Fig. 6A).
- ii) Compared analysis of branching coefficient in Lgr5 WT and KO organoids revealed differences between the genotypes depending on the Rspo type added to the culture medium. These differences were further confirmed by RNAseq studies performed on organoids (new Fig. 6C, 6D and new Fig. EV5C, EV5D).

Together, these ex vivo experiments suggest that Lgr5/Rspo2 interaction regulates Wnt activity in organoids and ECM production in the epithelial cells. These new findings are discussed in the Discussion section.

3. Based on Fig4D, the authors conclude that a conversion of autocrine Wnt ligand expression occurs from Wnt 5a in fetal intestines to Wnt 3 in adults. This is a surprising result given that previous reports have shown that Wnt5a is a mesenchymal factor and Wnt3 is expressed in Paneth cells. Why do the authors conclude that these are autocrine signals? The conclusions drawn from the transcriptional profiling of Lgr5GFP+ cells should be validated by ISH.

Our conclusion for autocrine production of Wnt ligands by ISCs is based on the data obtained from the FACS-isolated eGFP⁺ ISCs. Although expression of Wnt ligands is known to predominantly come from stromal cells, expression of Wnt5a in epithelial cells at this developmental stage was already suggested by a previous study (Nigmatullina L et al, 2017, EMBOJ). In the revised ms, we provide new RNAscope experiments showing epithelial expression of Wnt5a in E16.5 tissues (see new Fig. EV4D). Note that the former Fig 4D showing relative Wnt ligand expression levels are now presented in the new Supplementary Fig EV4C. Of note, the fact that Wnt2b, the main Wnt ligand released from subepithelial myofibroblasts, does not appear to be expressed in E16.5 eGFP isolated epithelial ISCs, further sustains the idea that Wnt ligand expression in our RNAseq data comes from an autocrine production.

Referee #3:

"Lgr5 Controls Extracellular Matrix Production By Stem Cells In the Developing Intestine", submitted by Vallone, V. et al, examines the functional consequence of inactivating Lgr5 on the development of the intestine. The authors find that inactivation of Lgr5 during development results in precocious Paneth cell differentiation, an increase in WNT tone, and expanded stem cell compartment. They further show that these phenotypes are dependent on WNT, as experimentally reducing Wnt signaling eliminates the difference between knock outs and controls. The authors further explore the functional

consequences of inactivating *Lgr5* using RNAseq and identify a difference in the *in vitro* response of two different LGR5 ligands, *Rspo1* and *Rspo2*.

Overall the paper is well written and easy to follow. The experiments are well performed and the findings of general interest. Although the Paneth and stem cell phenotype has previously been described, the authors provide additional mechanistic insight and demonstrate the dependence of the phenotype on WNT. The paper then shifts gears and reports on an RNAseq experiment and *in vitro* response to RSPOs. The findings of these experiments are quite interesting and novel, but **require additional follow up to assess their relevance**. As written, the manuscript does not have a single key finding; there are several interesting observations that are **not demonstrated to be connected**. I believe this can be improved with additional experiments.

As mentioned above, the new experiments performed on organoids with the various Rspodins connect *Lgr5* function with *Rspo2*/Wnt activity and ECM production. The whole text and discussion sections have been rewritten accordingly. We believe that cohesiveness of the whole ms has substantially been improved.

MAJOR:

1. The power of the RNAseq experiment is to determine the differences between control and LGR5 KO stem cells in embryos, in adults, over developmental time, and whether there is an interaction between genotype and developmental time. The manuscript would be improved if these data were presented **more concisely**. For example, two graphs (4B and 4E) are presented with some redundant data between the graphs. The authors could **use one graph** that would include differences between 16.5 HEvKO, Adult HEvKO, 16.5HE v Adult HE. This would make it easier to visualize the interaction between genotype and time that the authors suggest. Furthermore, the authors should explore statistical methods to use to assess the significance of this interaction.

As suggested, the two graphs from the ex Fig. 4 have been combined to a single one in the new Fig. 4B. This graph has been generated from *in silico* studies that intrinsically already rely on statistical methods (Degust followed by GSEA MolSig-Hallmark dataset).

Besides, in the new ms, data presented in the former Fig. 1 and Fig.5 (ex vivo cultures on *Rspo1*-containing medium) have been combined to the new Fig.1 describing the intestinal phenotype in *Lgr5*-deficient embryos.

2. The manuscript will be further improved by following up on what was identified in the RNAseq experiment. The authors **should include PCR, and tissue-based techniques, exploring the effects on ECM production and inflammatory response**. This should be done with **a developmental sequence in control and KOs**. Furthermore, it would really tie the manuscript together **to include embryos from LGK974 treated dams**. This will help assess whether the same processes that lead to precocious Paneth cell differentiation also affects ECM production and inflammatory responses.

To validate the RNAseq data regarding modulated expression levels of ECM/inflammation markers in progenitor cells versus adult epithelium, we have now added as new Fig.4D RNAscope experiments showing differential expression of some representative markers (*Col1a1*, *Zeb2* and *Cxcl10*) in embryos and adults (see also answer to Referee 1, point 1). Regarding embryonic tissues E18.5 treated with the porcupine inhibitor, qRT-PCR performed on whole tissues did not allow to draw any conclusions as most of the signal comes from the extra-epithelial compartment. Ideally, one would have needed to perform RNAseq on isolated eGFP⁺ve ISCs following LGK974 treatment, an experiment complicated by the fact that the total number of stem cells is reduced by the treatment (as demonstrated in the new Fig. 2C). Nevertheless, new RNAseq data obtained from new experiments performed on organoids (presented in the new Fig. 6C,6D and Fig. EV5C, EV5D) evidence a link between *Lgr5*/*Rspo2* interaction and ECM synthesis associated with high Wnt signaling activity and Paneth lineage differentiation. The detailed molecular mechanisms involved will deserve additional studies that appear beyond the scope of the present study.

3. The second novel observation regarding the differential effect of RSPO1 and RSPO2 in culture needs to be further explored. RSPO2 is much more potent than RSPO1 and has a dose-response curve that is an inverted U. Depending on what part of the curve one is assessing might lead to disparate conclusions. The authors need to **show a dose-response of both RSPO1 and RSPO2 using concentrations lower than 15 ng/ml**. For example, a curve of .1, 1, 10, 50, 100 might be a good place to start.

As suggested, new experiments have been performed to explore the impact of dose-response curves for organoid survival and morphology and these new data are now presented in the new Fig. 6. The activity of the Rspodins in this type of experiment is measured by survival curves. In our experiments, the inflexion point for Rspo1 and Rspo3 occurs between 100 and 15 ng/ml meanwhile it is observed between 5 and 0.8 ng/ml for Rspo2; indicating that in this system, Rspo2 is more active than the two other ligands. Note that we did not observe the suggested inverted U curve in our experimental conditions for the Rspo2 ligand. Therefore, for the new RNAseq studies, we have used as key concentration points of 100 ng/ml and 5 ng/ml for the 3 Rspodins. See also answer to specific point 2 from Referee 2.

MINOR:

1. The authors should provide a **close up of the intervillus region in Figure 2B**. It is very difficult to see any effects on beta-Catenin in the low power image.

As suggested, this panel has been replaced by a new Fig. 2C that also integrates quantification of the Olfm4⁺ ISCs in fetal tissues. See also answer to Referee 1, point 3.

2. How are Paneth cells defined in Figure 2C? Can the authors **include PCR data on additional Paneth cell markers**, not just Crypt5? It would be nice to include the same markers that were used to describe the phenotype in Figure 1.

In embryos, Paneth cell differentiation is measured by Lendrum staining (based on visualization of fuschia-stained defending granules, as defensins are the earliest markers for differentiation into this cell lineage). Indeed, Paneth cell differentiation normally occurs in the postnatal stage and expression of Plyz, classically used for quantification of these cells in adult tissues, has not reached sufficient levels to be detected in E18.5 Lgr5 KO embryos at the protein level meanwhile Lyz1 transcripts can be detected by qRT-PCR (as in the new Fig. 1B). As suggested, we have added qRT-PCR data with additional Paneth markers in the new supplementary Fig. EV2D. Of note, the effect of Wnt inhibition on their expression is not as sensitive as for the Defa5 gene, this might be related to much lower expression level variations for these additional markers.

3. It is unclear why two different periods of chase were used in early PN versus adult animals in Figure 3. The authors should be consistent and use a **10 day chase in the adults**. Longer chases may be confounded by differences in Paneth cell lifespan.

As suggested, we have performed a new lineage tracing experiment on adults using the same chase of 10 days. These data are now added in a new Fig. 3B. As observed initially with a chase of 1 month, no overt changes in clone tracing is detected in Lgr5 cKO vs Lgr5-HEs, coherent with no overt phenotype related to Lgr5 loss in ISCs under homeostasis. Moreover, the number of newly generated Plyz⁺ cells in recombined crypts is not affected in homeostatic adult tissues (data presented in a new Fig. 3C).

4. It may be possible to make the **manuscript more concise** by combining Figures 1 and 5. In addition, Figure 4D can be moved to Extended Data and this figure can be simplified by combining graphs B and E.

As recommended, we have combined data from ex Fig.1 and Fig. 5 in the new Fig. 5. Note that qRT-PCR formerly presented in ex fig. 5C have been removed to avoid data redundancy with new Fig. 6. In silico analyses formerly presented in Fig. 4B and 4E are combined into a new Fig. 4B (see also answer to Referee 1 point 1).

Data initially presented in Fig. 4D have been dispatched at follows:

- RNAscope experiments showing Axin2 expression in Lgr5 WT and KO E16.5 intestines have been replaced by RNAscope experiments showing Axin2 expression in Lgr5 WT and KO E18.5 intestines in the new Fig. 2D
- FACS analysis showing amplification of the stem cell pool has been moved to new Fig 1E,
- Graphs showing relative expression levels for Wnt ligands during development and in Lgr5 Ko versus HE ISCs at E16.5 have been moved to Fig. EV5C (see also answer to ref 2 point 3).

Dear Dr. Garcia,

Thank you for the submission of your revised manuscript to our editorial offices. We have now received the reports from the three referees that were asked to re-evaluate your study, you will find below. As you will see, the referees now support the publication of your study in EMBO reports. Nevertheless, all three referees have remaining concerns, mostly text revisions, I ask you to address in a final revised version of the manuscript. Please also provide a point-by-point response that addresses the remaining points of the referees.

Further, I have these editorial requests I ask you to address:

- Please provide the abstract written in present tense.

- Please format the references according to our journal style. Please list not more than 10 authors.

See:

- Please add call-outs for the four datasets EV1-4. There is one call-out for a table S1. Please check. We also need legends for the 4 datasets. Please add a legend explaining the content on the first TAB of the excel files for these datasets.

- Please also add a formal "Data Availability section" to the manuscript after the methods section. This is now mandatory (like the COI statement). If no primary datasets have been deposited in any database, please state this in this section (e.g. 'No primary datasets have been generated and deposited').

The accession numbers and database should be listed in a formal "Data Availability " section (placed after Materials & Methods) that follows the model below. Please note that the Data Availability Section is restricted to new primary data that are part of this study.

Data availability

- RNA-Seq data: Gene Expression Omnibus GSE46843

(<https://www.ncbi.nlm.nih.gov/geo/query/acc.cgi?acc=GSE46843>)

- [data type]: [name of the resource] [accession number/identifier/doi] ([URL or identifiers.org/DATABASE:ACCESSION])

- Please add scale bars to all microscopic images. Figs. EV1B and 2B presently have no scale bars. Do not write on the scale bars, and provide their length only in the respective figure legend.

- Regarding data quantification and statistics, can you please check again that, where applicable,

the number "n" for how many independent experiments (biological vs. technical replicates) were performed, the bars and error bars (e.g. SEM, SD) and the test used to calculate p-values is specified in the respective figure legends.

- Please enter all the funding information also into our submission system. Please check that in the online form and the manuscript the funding information is the same and complete.

- Finally, please find attached a word file of the manuscript text (provided by our publisher) with changes we ask you to include in your final manuscript text, and some queries, we ask you to address. Please provide your final manuscript file with track changes, in order that we can see the modifications done.

In addition I would need from you:

- a short, two-sentence summary of the manuscript
- two to three bullet points highlighting the key findings of your study
- a schematic summary figure (in jpeg or tiff format with the exact width of 550 pixels and a height of not more than 400 pixels) that can be used as a visual synopsis on our website.

Yours sincerely

Achim Breiling
Editor
EMBO Reports

Referee #1:

Major comments:

1. The authors nicely showed that the stem cell/Paneth cell prematuration phenotype in Lgr5 KO occurs through effects on Wnt signaling. In the initial figures, increased expression of Wnt and differentiating Paneth cells were positively correlated with increased branching morphogenesis (Fig 1-3). However, in Fig 6B, the degree of branching was reduced, despite increased Wnt signaling and Paneth cells. A clearer explanation regarding the hypothesis that the authors have to describe the observation would be helpful.

2. Related to the above comment, the authors suggested that the Rspo2/Lgr5 interaction negatively regulates Paneth cell differentiation (which is induced by higher Wnt). However with 5ng/mL Rspo2, Lgr5 KO shows increased Paneth cells in comparison to WT control. A clearer explanation of the relationship between Rspo2 levels and Wnt signaling or Paneth cell number (with a diagram showing the relationship between the different components, perhaps) may help readers.

Minor comments:

1. Fig 4A: The rest of the data presented in the study are in order of WT, HE, KO except for that heat map. Suggest switching the order.
2. Fig 4B GSEA analysis panels: Is it possible to make the labels in the plot bigger so that they are legible?
3. Line 193: Text mentions qPCR experiments. However, the data do not seem to be provided in the

figures.

Referee #2:

In summary, this manuscript demonstrates that Lgr5 negatively regulates Wnt-mediated ISC formation and Paneth cell differentiation during gut development. Transcriptional profiling also showed that loss of Lgr5 is associated with downregulation of numerous ECM components. The authors conclude the study by examining the role of Rspo2 in mediating Lgr5-dependent effects. The experiments presented in the revised manuscript address many of the comments which were raised in the first submission. However, the major concern that I have with this manuscript is in the interpretation of the data. Indeed, their conclusions are very often unclear and/or unjustified. The lack of coherence of the study, an issue raised by the other reviewers, remains an additional point of concern. In its current form, this manuscript is not suitable for publication in EMBO Reports.

Major comments:

In the original manuscript the authors made the interesting assertion that Rspo2 and Rspo1 play non-redundant roles. The authors noted that Rspo2 downregulated Wnt /ISC markers when compared to Rspo1 in a Lgr5-dependent fashion. These findings led the authors to conclude that Rspo2/Lgr5 receptor interactions negatively regulate the Wnt/ β -catenin pathway in ISCs. Surprisingly, in the revised manuscript the authors now show that Rspo2 strongly activates ISC markers compared to Rspo1/3 and that loss of Lgr5 further potentiates the ability of Rspo2 to drive ISC marker expression. Rspo2 was also by far the most potent stimulator of organoid growth compared to all other Rspodins. Based on these findings one may conclude that Lgr5 acts as negative regulator specifically of Rspo2 function. Disappointingly the authors do not really discuss these observations except to state that their findings are "...providing evidences that the Lgr5/Rspo2 interaction in ISCs regulates the ISCs number by modulating the Wnt tone and ECM production in the epithelium." The concept of Wnt tone is completely unclear in this context and should be further defined. Similarly, in the abstract the authors state that Lgr5/Rspodin 2 interaction negatively regulates the pool of ISCs in organoids. Again, this statement is very misleading. In my view the authors can only state that Lgr5 antagonizes the growth stimulatory functions of Rspo2. They also claim that their findings are consistent with previous reports suggesting that Rspo2 is a tumor suppressor. Rspo2 is clearly not suppressing ISCs based on the new evidence shown in Fig 6. This part of the manuscript must be revised.

The authors demonstrate that Lgr5 deficiency leads to decreased expression of genes encoding for ECM components. Based on these results the authors conclude that "...Lgr5 controls fetal ISC maturation associated with acquisition of a definitive stable epithelial phenotype, that depends on the capacity of ISCs to generate their own extracellular matrix." The authors provide no evidence to suggest that maturation of fetal ISCs is dependent on their ability to produce their own ECM. Compared to the stroma the contribution of ECM components by the epithelium is almost negligible during fetal development. The decrease in ECM related genes in the epithelium may only be a marker of maturation. Whether the downregulation of ECM causes ISC maturation remains completely unknown.

In general, I would recommend the authors completely rewrite the manuscript.

Minor comments

In response to my comments about the expression of Rspodins, the authors performed RNAScope

experiments in Figure 5. As expected, the authors found prominent expression of Rspo 2 and 3 in stromal cells. Based on the images shown expression of Rspo1-3 was also detected in the epithelium at all stages. These findings should be at least commented on in the manuscript. Also, the histology of the adult tissue is of poor quality and these stainings should be repeated.

Referee #3:

Fernandez-Vallone et al have resubmitted a substantially improved manuscript. The organization of the data is simplified and unified and results presented in a more coherent manner.

The majority of my concerns/suggestions have been addressed. There are some remaining questions.

1. It was suggested to perform tissue-based techniques on developing WT and KO animals to validate the RNAseq results. The authors have nicely included these data for the developmental sequence, but did not include KO animals. This is a notable absence and has left me puzzling why. Likely, it is because the epithelial expression of these genes is quite low, making it difficult to measure a further decrease in KOs. If this is the case, perhaps the authors should mention it. However, this raises an additional question. Why is the epithelial expression so low? Is it possible there is mesenchymal contamination of the FACS isolated GFP cells, or a possibility that there are Lgr5 GFP positive mesenchymal cells that are influencing the RNAseq results? If the authors do not want to include data on these possibilities, they may want to include statements addressing them.

2. The RSPO data are much improved. The data on RSPO2 look like it is more potent in the absence of LGR5. It is hard to conclude because the data below 0.8 ng/ml have not been included. However, if true, it would suggest that LGR5 acts as an inhibitor of RSPO2 response, a very interesting idea. The authors may want to mention this, and perhaps tie it to RSPO-LGR5 affinities e.g. are there existing data that RSPO2 has a higher affinity for LGR5 than the other RSPOs?

3. In Figure 4B, according to the legend there should be 3 bars (red, blue, green) in each of the bins. Several of the bins have more than 3 bars, which I don't understand. Perhaps there is some graphing mistake?

Overall, an interesting and thought provoking set of experiments.

Referee #1:

Major comments:

1. *The authors nicely showed that the stem cell/Paneth cell prematuration phenotype in Lgr5 KO occurs through effects on Wnt signaling. In the initial figures, increased expression of Wnt and differentiating Paneth cells were positively correlated with increased branching morphogenesis (Fig 1-3). However, in Fig 6B, the degree of branching was reduced, despite increased Wnt signaling and Paneth cells. A clearer explanation regarding the hypothesis that the authors have to describe the observation would be helpful.*

The experiments performed in Rspo1 100 ng/ml-culture conditions, presented in Fig. 1F/EV1C and Fig. 6 of the former version, are not completely comparable. Indeed, branching coefficient studies reported in Fig 1/EV1 were done on samples freshly isolated from tissues (designated as “initial seeding experiments”) in which virtually no Paneth cells (PC) were yet present in WTs meanwhile KOs already contained matured PC. It is likely that in this case, as published earlier, PC in KO samples significantly increase plating efficiency and organoid complexity as compared to WTs (see Sato et al, 2009-reference 14 in the ms). Experiments presented in Fig. 6 with Rspo1 100 ng/ml were done on “replated organoids”, a situation in which PC had been generated in WTs samples due to high stimulatory conditions as compared to the *in vivo* embryonic intestine. In this situation, no statistically significant difference was observed between WTs and KOs. In contrast, performing the same experiments with the same “replated organoid samples” but in presence of Rspo2-5 ng/ml (instead of Rspo1-100 ng/ml), induced a clear statistical reduction in branching coefficient in KOs as compared to WTs.

In order to facilitate data interpretation of organoid branching coefficient studies, we have now introduced a graphical scheme in Fig. 1F and Fig. 1G, Fig. 6, Fig. EV1C and EV1D/EV1E to better visualize experiments performed on “initial” or “replated” samples. In addition, we have modified the Results section by further describing results and their interpretation (lines 98-100 in page 5; lines 207-215 in pages 9-10). Moreover, we have further discussed organoid experiments (Rspodin ligands or/and Lgr5 effects) in the Discussion section (lines 289-304, pages 12-13).

2. *Related to the above comment, the authors suggested that the Rspo2/Lgr5 interaction negatively regulates Paneth cell differentiation (which is induced by higher Wnt). However with 5ng/mL Rspo2, Lgr5 KO shows increased Paneth cells in comparison to WT control. A clearer explanation of the relationship between Rspo2 levels and Wnt signaling or Paneth cell number (with a diagram showing the relationship between the different components, perhaps) may help readers.*

As suggested, to clarify our interpretation of the possible relationship between Rspodin ligands (including Rspo2), Lgr5 receptor, Wnt signaling and PC differentiation, we have now added a tentative working model hypothesis in the new Fig. EV5F. In addition, we have further explained the data

obtained in organoid experiments from Fig. 6 in the new Results section (lines 207-215 in pages 9-10). and detailed data interpretation in the Discussion section (lines 289-304, pages 12-13).

To summarize, experiments presented in Fig. 6 suggest that any Rspodin ligand, (including Rspo2, but less efficiently Rspo4) potentiates Wnt activity in WT ISCs in a dose-dependent manner (100 versus 15 versus 5 ng/ml): higher concentration of ligand correlated with higher organoid survival and higher Wnt activity (Fig. 6 A to D). Note that by “Wnt activation” we meant stimulation of the expression of Wnt target genes, expression of ISCs markers and differentiation towards the PC lineage. Similar Rspo1 and Rspo3-dose dependent effects on organoid survival and Wnt activity were detected on Lgr5 KO organoids. In contrast, the Rspo2 culture condition maintained high Wnt activity in absence of the Lgr5 receptor in Lgr5 KO organoids. Taken altogether, these data would imply that:

1. ISCs express receptors other than the sole Lgr5, able to bind Rspodins, including the Rspo2 ligand; such interaction results in potentiation of Wnt activity in ISCs and organoid survival. As already discussed in the former version of the ms, these receptors could be the paralogue Lgr4, but also Syndecans and/or Glypicans, detected to be significantly expressed in ISCs (now Fig. EV4C).
2. In presence of Lgr5, the Wnt potentiation effect mediated by Rspo2 through the other receptors would be attenuated. Such “buffering effect” being lost in Lgr5-deficient ISCs, this would result in the observed increased Wnt activity and bias towards PC differentiation.

Minor comments:

1. Fig 4A: The rest of the data presented in the study are in order of WT, HE, KO except for that heat map. Suggest switching the order.

The revised ms integrates this request in the new Figure 4 panel 4A.

2. Fig 4B GSEA analysis panels: Is it possible to make the labels in the plot bigger so that they are legible?

The revised ms integrates this request in the new Figure 4 panel 4B.

3. Line 193: Text mentions qPCR experiments. However, the data do not seem to be provided in the figures.

We deeply apologize for this mistake, we omitted to introduce Rspodin expression levels measured by q-RT-PCR on whole tissues in the former revised V2 ms. We have now added these data in the new Fig. EV5A (with Call-out in the text, line 199, page 9). Ct values confirm the RNAscope studies indicating that all Rspodins are detected to be expressed at low levels in the developing and adult intestine, and that the ligands Rspo2 and Rspo3 are the most expressed ones in the stromal compartment of the intestine.

Referee #2:

*In summary, this manuscript demonstrates that Lgr5 negatively regulates Wnt-mediated ISC formation and Paneth cell differentiation during gut development. Transcriptional profiling also showed that loss of Lgr5 is associated with downregulation of numerous ECM components. The authors conclude the study by examining the role of Rspo2 in mediating Lgr5-dependent effects. The experiments presented in the revised manuscript address many of the comments which were raised in the first submission. However, the major concern that I have with this manuscript is in the interpretation of the data. Indeed, their **conclusions are very often unclear and/or unjustified. The lack of coherence of the study, an issue raised by the other reviewers, remains an additional point of concern.** In its current form, this manuscript is not suitable for publication in EMBO Reports.*

We agree that the lack of coherence of the version V1 of the paper was raised as a general comment by all referees. In the version V2, we have extensively revised the text and figures to improve the quality of the ms and we appreciate that this has been positively noticed by Referee 3.

Regarding the Referee 's feeling for "unclear and/or unjustified conclusions", this might have resulted from insufficient explanation in the former ms version V2 of the organoids experiments. In the revised V2 ms, we have further explicated and discussed data and their interpretation and provide a working model hypothesis in the new Fig. EV5F (see below).

Major comments:

In the original manuscript the authors made the interesting assertion that Rspo2 and Rspo1 play non-redundant roles. The authors noted that Rspo2 downregulated Wnt /ISC markers when compared to Rspo1 in a Lgr5-dependent fashion. These findings led the authors to conclude that Rspo2/Lgr5 receptor interactions negatively regulate the Wnt/ β -catenin pathway in ISCs. Surprisingly, in the revised manuscript the authors now show that Rspo2 strongly activates ISC markers compared to Rspo1/3 and that loss of Lgr5 further potentiates the ability of Rspo2 to drive ISC marker expression.

We respectfully disagree with this comment. There is neither discordance between ms version V1 and V2 nor with new data interpretation. In the version V1, we showed that Rspo2 ligand sustained organoid growth and in the version V2, we obtained further evidences not only that Rspo2 is potentiating Wnt activity in a dose-dependent manner, likely mediated by other receptors in WT organoids (see former text lines 272-278); but also that this signal is further accentuated in absence of the Lgr5 receptor; Lgr5 would therefore behave as a negative regulator of the Wnt pathway in WT ISCs via its interaction with Rspo2.

However, we agree that misunderstanding of our interpretation could be due to insufficient explanation of the results, which involve two different notions: differences between ligand effects and differences between receptor effects. In order to further clarify the ms, we have revised Results (lines 207-227 in pages 9-10) and Discussion sections (lines 289-304, pages 12-13). See also Answer to Referee 1 major comments 1/2.

Rspo2 was also by far the most potent stimulator of organoid growth compared to all other Rspodins. Based on these findings one may conclude that Lgr5 acts as negative regulator specifically of Rspo2 function. Disappointingly the authors do not really discuss these observations except to state that their findings are "...providing evidences that the Lgr5/Rspo2 interaction in ISCs regulates the ISCs number by modulating the Wnt tone and ECM production in the epithelium." The concept of Wnt tone is completely unclear in this context and should be further defined.

First of all, as it is generally accepted by the research community, the Wnt tone was defined by evidences for expression in organoids of known Wnt target genes, stem cell markers and PC markers (such as Axin2, Ascl2, Tnrfsf19, Lgr5, Olfm4, CD44, Defa genes). This point is now explicated in the results section (lines 220-221, page 10). Second, the conclusion that "the Lgr5/Rspo2 interaction in ISCs regulates the ISCs number" is based on data presented in Fig. 6E showing that the number of Olfm4^{+ve} cells is reduced in WT as compared to Lgr5 KO organoids when grown in presence of Rspo2 (lines 225-227, page 10). To clarify our point of view, we have further discussed about the Lgr5/Rspo2 interaction effect on Wnt activity in the new version V2 of the ms (lines 289-304, page 13).

Similarly, in the abstract the authors state that Lgr5/Rspodin 2 interaction negatively regulates the pool of ISCs in organoids. Again, this statement is very misleading. In my view the authors can only state that Lgr5 antagonizes the growth stimulatory functions of Rspo2.

See comment above. The conclusion that "the Lgr5/Rspo2 interaction in ISCs regulates the ISCs number", and thus the pool of ISCs, is based on data presented in Fig. 6E showing that the number of Olfm4^{+ve} cells is reduced in WT as compared to Lgr5 KO organoids when grown in presence of

Rspo2. In addition, based on RNAseq and qRT-PCR experiments provided in the new Fig. 6C-D and new Fig. EV5B and EV5D, presence or absence of Lgr5 also modifies the differentiation potential of ISCs when organoids are cultured in presence of Rspo2 (a bias towards PC differentiation being observed in absence of Lgr5).

As suggested by referee, we have explicated in the new text "Together, these data suggest that the Lgr5 receptor acts as an inhibitor of the Rspo2 response in ISCs" (lines 228-229, page 10). The abstract has also been modified with the new sentence "Finally, using the *ex vivo* culture system, evidences were provided that Lgr5 antagonizes the Rspodin 2-Wnt mediated response in ISCs in organoids, revealing a sophisticated regulatory process for Wnt signaling in ISC." (lines 26-28, page 2).

They also claim that their findings are consistent with previous reports suggesting that Rspo2 is a tumor suppressor. Rspo2 is clearly not suppressing ISCs based on the new evidence shown in Fig 6. This part of the manuscript must be revised.

From our results, we concluded that Rspo2 (but not Rspo1 or Rspo3 ligands) negatively regulates the pool of ISCs via its interaction through the Lgr5 receptor (but not the other receptors). As written in the former V2 version "These data are in line with a previous report showing that Rspo2, but not other family members, exhibits tumor suppressive activity on colorectal cancer cell lines *via* negative regulation of the Wnt/ β -catenin pathway through interaction with the Lgr5 receptor [20]". We cited this paper because it mentions that the specific negative regulation of Wnt activity occurs via the Rspo2/Lgr5 interaction. It is clear that the citation was not related to the "tumor suppressor notion", which cannot be claimed from data presented in Fig. 6 dealing with normal (non-tumoral) ISCs.

The authors demonstrate that Lgr5 deficiency leads to decreased expression of genes encoding for ECM components. Based on these results the authors conclude that "...Lgr5 controls fetal ISC maturation associated with acquisition of a definitive stable epithelial phenotype, that depends on the capacity of ISCs to generate their own extracellular matrix." The authors provide no evidence to suggest that maturation of fetal ISCs is dependent on their ability to produce their own ECM. Compared to the stroma the contribution of ECM components by the epithelium is almost negligible during fetal development. The decrease in ECM related genes in the epithelium may only be a marker of maturation. Whether the downregulation of ECM causes ISC maturation remains completely unknown.

We did not mean that ECM production in epithelium is more contributive than the stromal one to fetal development. But, we provide evidences that ECM production in ISCs is associated with changes in ISCs maturation (based on the RNaseq data obtained at E16.5 Lgr5 KO vs WTs; compared ISCs at E18.5 and adult stages). But, we agree that we did not provide direct evidences for a causal effect in the present ms (i.e. loss of ECM causing ISCs maturation). We have changed this sentence in the abstract by "Moreover, transcriptome analyses reveal that Lgr5 controls fetal ISC maturation associated with acquisition of a definitive stable epithelial phenotype, *as well as* the capacity of ISCs to generate their own extracellular matrix. (lines 24-26, page 2, and lines 69-70, page 4).

In general, I would recommend the authors completely rewrite the manuscript.

As suggested and commented above, we have rephrased the indicated parts in Results and Discussion sections.

Minor comments

In response to my comments about the expression of Rspodins, the authors performed RNAscope experiments in Figure 5. As expected, the authors found prominent expression of Rspo 2 and 3 in stromal cells.

We performed these experiments as suggested by the referee because they could be informative. As far as we know, no reports were available regarding Rspodin ligand expression in the developing intestine. In our view, one cannot “**expect**” what will be these future results until they are generated.

*Based on the images shown expression of Rspo1-3 was also detected in the epithelium at all stages. **These findings should be at least commented on in the manuscript.** Also, the **histology of the adult tissue is of poor quality** and these stainings should be repeated.*

As explained above (Referee 1, minor point 3), we omitted to present some qPCR experiments on tissue expression of Rspodins in the former V2 version of the ms. We have now added these results in the new Fig. EV5A. They confirm RNAscope data. In addition, as suggested, we have also added a sentence to comment on the very low expression of ligands in the epithelium in the Results section (lines 199-202 in page 9). Regarding the quality of the adult pictures in Fig. 5, we agree that the Rspo1-Rspo2/adult images were slightly scanned out of focus. We provide now new focused pictures.

Referee #3:

Fernandez-Vallone et al have resubmitted a substantially improved manuscript. The organization of the data is simplified and unified and results presented in a more coherent manner. The majority of my concerns/suggestions have been addressed. There are some remaining questions.

*1. It was suggested to perform tissue-based techniques on developing WT and KO animals to validate the RNAseq results. The authors have nicely included these data for the developmental sequence, but **did not include KO animals.** This is a notable absence and has left me puzzling why. Likely, it is because the epithelial expression of these genes is quite low, making it difficult to measure a further decrease in KOs. If this is the case, **perhaps the authors should mention it.***

It is true that we could not verify by RNAscope the data generated from RNAseq for Lgr5 E16.5 KOs vs WTs, likely due to the low expression level of the tested genes in the epithelium. A sentence has been added to mention this point (lines 162-163, page 8).

*However, this raises an additional question. Why is the epithelial expression so low? Is it possible there is **mesenchymal contamination** of the FACS isolated GFP cells, or a possibility that there are **Lgr5 GFP positive mesenchymal cells** that are influencing the RNAseq results? If the authors do not want to include data on these possibilities, they may want to **include statements addressing them.***

Regarding the raised potential contamination of epithelial GFP^{+ve} ISCs with any other kind of mesenchymal cell, if so, the only explanation for such event would be that Lgr5 E16.5 HEs samples would be more contaminated than the Lgr5 KO ones, though having used the same method at the same time to isolate and sort the GFP^{+ve} cells. Although we cannot formally rule out this possibility, the risk appears rather low after having excluded potential doublets by the same gating strategy (using FSC-H/FSC-A followed by SSC-H/SSC-A parameters). Detailed gating strategy is now added as the new Fig. EV4A and referred to in the Results section (lines 150-151, page 7). Regarding the potential contamination of epithelial GFP^{+ve} ISCs by mesenchymal Lgr5^{+ve} cells, by the RNAscope method, it is true there are some GFP^{+ve} cells detected in the tip of the villi of adult tissues but we didn't detect any kind of mesenchymal Lgr5^{+ve} signal in the nascent villi or in stroma at the E16.5 stage (time point used for FACS sorting). These data are provided for referees and editor as Annex information (end of the point-by-point response). Note that for the sorting procedure in adults, single cells were obtained from enriched intestinal crypts (EDTA 5 mM-PBS treatment) after an initial scratching villi removal, a procedure that strongly reduces the risk of mesenchymal GFP^{+ve} cells contamination. As the data describing adult mesenchymal Lgr5^{+ve} cells in intestine are not yet published, and as we do not want to scoop this work from another group, we would prefer not to mention this point in the ms.

2. The RSPO data are much improved. The data on RSPO2 look like it is more potent in the absence of LGR5. It is hard to conclude because the data below 0.8 ng/ml have not been included. However, if true, it would suggest that LGR5 acts as an inhibitor of RSPO2 response, a very interesting idea. The authors may want to mention this, and perhaps tie it to RSPO-LGR5 affinities e.g. are there existing data that RSPO2 has a higher affinity for LGR5 than the other RSPOs?

As observed in Fig. 6B, 0.8 ng/ml was not compatible with 100% survival for any of the tested ligands, including Rspo2 (Fig. 6B, left panel). For qRT-PCR, we did not get enough RNA to run in parallel all samples, including the ones coming from 0.8 ng/ml wells; this is the reason why this time point is not present in the qRT-PCR experiments presented in Fig. 6D. Our results are indeed in favor of a role for Lgr5 as an inhibitor of the Rspo2 activity. Considering also other referees' comments, we have further described these results (lines 207-215 and 218-229, pages 9-10) and discussed them in the new ms (lines 289-304, pages 12-13). A report from Carmon et al. (2011) indicated that the Rspo2 is the ligand showing the highest affinity to the Lgr5 receptor. This notion has also been discussed (lines 295-306, page 13). Note that, as suggested, we have now added as new Fig EV5F a working model hypothesis regarding proposed Rspodins/receptors interactions in ISCs.

3. In Figure 4B, according to the legend there should be 3 bars (red, blue, green) in each of the bins. Several of the bins have more than 3 bars, which I don't understand. Perhaps there is some graphing mistake?

The figure has been generated using the *in silico* analysis GSEA MolSig. In few cases, for instance "the KRAS signaling" Hallmark, among the differentially expressed genes some are upregulated and other are downregulated by this pathway, this all over indicating that this pathway is modulated differentially. We have mentioned this point in the Figure legends of the new Fig. 4B.

Annexes for Editor and Referees

Referee 3, major comment 1

RNAscope for Lgr5 expression in embryonic and adult intestine.

The presence of Lgr5+ve cells in the adult mesenchyme were detected in the tip of the villi (arrowheads), which are mechanically removed before the crypt enriched procedure and then single cell generation for FACS sorting of ISCs.

Referee 1, major comment 2

Working model hypothesis regarding the interactions of Rspodin ligands and their receptors in ISCs.

New Fig. EV5F

Dear Dr. Garcia,

Thank you for the submission of your revised manuscript to our editorial offices. We have now received the reports from the referees that were asked to re-evaluate your study, you will find below. As you will see, the referees now support the publication of your study in EMBO reports. Nevertheless, referee #2 has 2 remaining points I ask you to address in a further revised version of the manuscript. Both can be addressed by text changes. Please discuss the branching data in Fig. 6 as indicated by the referee, and describe how organoid branching was calculated in the Materials and Methods section. Please also provide a final response (point-by-point response) that addresses the remaining point of the referee regarding the branching data in Fig. 6.

Further, the writing in the synopsis image provided is rather small. See the attached version in the size the image will go online. Could you please upload me a version with bigger fonts in jpeg or tiff format with the exact width of 550 pixels and a height of not more than 400 pixels)?

Moreover, in the references journal names should be italicised. Please change that.

Kind regards,

Achim

Achim Breiling
Editor
EMBO Reports

Referee #1:

The authors have addressed my concerns from the previous review.

Referee #2:

With the most recent revisions of the text, the authors have significantly improved on the general quality of the manuscript. I only have one final comment which likely can be addressed with further modifications to the text.

In figure 6B, the authors quantified the branching coefficient of Lgr5 wt and KO organoids in the presence of various Rspodins. The authors showed significant reduction in organoid branching in Lgr5 KOs treated with Rspo2. This implies that Rspo2 positively regulates crypt morphogenesis or branching in a Lgr5-dependent fashion. If this is correct then how do these findings relate to the other figures showing that Lgr5 negatively suppresses Wnt signaling in a Rspo2 dependent fashion. Are the authors suggesting that high Wnt activity suppresses crypt branching? This would be somewhat counterintuitive. Or perhaps these findings reflect a Wnt independent function for Lgr5/Rspo2 that controls the number of crypt buds formed during organoid expansion. These

observations should be fully discussed.

The authors should also describe how organoid branching was calculated in the Materials and Methods section.

Referee #3:

The authors have adequately addressed my comments and have submitted a substantially improved manuscript.

Referee #2:

*With the most recent revisions of the text, the authors have significantly improved on the general quality of the manuscript. I only have one final comment which likely can be **addressed with further modifications to the text.***

1. In figure 6B, the authors quantified the branching coefficient of Lgr5 wt and KO organoids in the presence of various Rspodins. The authors showed significant reduction in organoid branching in Lgr5 KOs treated with Rspo2. This implies that Rspo2 positively regulates crypt morphogenesis or branching in a Lgr5-dependent fashion.

This conclusion point has now been explicitly introduced in the text Results section (lines 215-216 page 10).

*If this is correct then how do these findings relate to the other figures showing that Lgr5 negatively suppresses Wnt signaling in a Rspo2 dependent fashion. **Are the authors suggesting that high Wnt activity suppresses crypt branching? This would be somewhat counterintuitive.** Or perhaps these findings reflect a **Wnt independent function for Lgr5/Rspo2** that controls the number of crypt buds formed during organoid expansion. These observations should be fully discussed.*

We respectfully disagree and do not think that it would be counterintuitive. Indeed, it is well known that organoids grown in presence of Wnt3a or organoids deficient for Apc exhibit enhanced Wnt signaling activity, which translates in reduced branching of organoids that adopt a round cystic morphology. To clarify this point regarding organoid branching and Wnt signaling, we have discussed this notion in the new Results section of the revised V3 text (lines 230-233 page 10). On the other hand, the potential involvement of Wnt-independent pathways on crypt morphogenesis via the Lgr5/Rspo2 interaction is not excluded. Indeed, our RNAseq data from sorted E16.5 CBCs suggest that other pathways are deregulated in Lgr5-deficient stem cells as compared to Lgr5 HEs (though, we do not have evidences that this is directly involving Rspo2 ligand). A sentence regarding putative Wnt-independent functions mediated by Lgr5/Rspo2 axis on organoid branching has now been added in the text (lines 233-234, page 10).

2. *The authors should also describe how organoid branching was calculated in the Materials and Methods section.*

In the former versions of the manuscript, the formula used to calculate branching coefficient was reported directly on the Figures Fig. 6B right panel and Fig. EV1C : the y axis of graphs showed branching coefficient as “ $1-4\pi(\text{area}/\text{perimeter}^2)$ ”. As suggested, this is now also reported in the text of Materials and methods in the “Ex vivo culture” paragraph (lines 360-363 page 16) and a new reference describing how to calculate branching coefficient has been added as the reference 45.

Dr. Marie-Isabelle Garcia
Université Libre de Bruxelles
Inst. de Recherche Interdisciplinaire (IRIBHM).
route de Lennik 808
Brussels B-1070 Brussels
Belgium

Dear Dr. Garcia,

I am very pleased to accept your manuscript for publication in the next available issue of EMBO reports. Thank you for your contribution to our journal.

At the end of this email I include important information about how to proceed. Please ensure that you take the time to read the information and complete and return the necessary forms to allow us to publish your manuscript as quickly as possible.

As part of the EMBO publication's Transparent Editorial Process, EMBO reports publishes online a Review Process File to accompany accepted manuscripts. As you are aware, this File will be published in conjunction with your paper and will include the referee reports, your point-by-point response and all pertinent correspondence relating to the manuscript.

If you do NOT want this File to be published, please inform the editorial office within 2 days, if you have not done so already, otherwise the File will be published by default [contact: emboreports@embo.org]. If you do opt out, the Review Process File link will point to the following statement: "No Review Process File is available with this article, as the authors have chosen not to make the review process public in this case."

Should you be planning a Press Release on your article, please get in contact with emboreports@wiley.com as early as possible, in order to coordinate publication and release dates.

Thank you again for your contribution to EMBO reports and congratulations on a successful publication. Please consider us again in the future for your most exciting work.

Yours sincerely,

Achim Breiling
Editor
EMBO Reports

THINGS TO DO NOW:

You will receive proofs by e-mail approximately 2-3 weeks after all relevant files have been sent to

our Production Office; you should return your corrections within 2 days of receiving the proofs. Please inform us if there is likely to be any difficulty in reaching you at the above address at that time. Failure to meet our deadlines may result in a delay of publication, or publication without your corrections. All further communications concerning your paper should quote reference number EMBOR-2019-49224V4 and be addressed to emboreports@wiley.com.

If sequence, structural, or microarray data is included, the data must be deposited with the appropriate databases. Please follow our instructions to authors, which can be found at: http://embor.msubmit.net/html/embor_author_instructions.html.

Should you be planning a Press Release on your article, please get in contact with emboreports@wiley.com as early as possible, in order to coordinate publication and release dates.

IMPORTANT

We will only be able to proceed with publication if you have completed and signed the correct License to Publish form and the Page Charge Authorization form (this charge is applicable to Scientific Reports and Articles only):

- PAGE CHARGE AUTHORISATION (For Scientific Reports and Articles only)
<https://www.embopress.org/page/journal/14693178/authorguide#chargesguide>

- LICENSE TO PUBLISH

Once your article has been received by Wiley for production the corresponding author will receive an email from Wiley's Author Services system which will ask them to log in and will present them with the appropriate license for completion.

OPEN ACCESS papers

Authors of accepted peer-reviewed original research articles may choose to pay a fee in order for their published article to be made freely accessible to all online immediately upon publication. The EMBO Open fee is fixed at \$5,200 (+ VAT where applicable).

We offer two licenses for Open Access papers, CC-BY and CC-BY-NC-ND.

For more information on these licenses, please visit: <http://creativecommons.org/licenses/by/3.0/> and http://creativecommons.org/licenses/by-nc-nd/3.0/deed.en_US

- PAYMENT FOR OPEN ACCESS papers

You also need to complete our payment system for Open Access articles. Please follow this link and select EMBO Reports from the drop down list and then complete the payment process: https://authorservices.wiley.com/bauthor/onlineopen_order.asp

- ORCID

EMBO Press encourages all authors and reviewers to associate an Open Researcher and Contributor Identifier (ORCID) to their account. ORCID is a community-based initiative that provides an open, non-proprietary and transparent registry of unique identifiers to help disambiguate research contributions. The Corresponding Author must provide his or her ORCID ID.

Corresponding Author Name: MJ RUST & AR DINNER

Journal Submitted to: MSB

Manuscript Number: MSB-19-9355